# Quantifying noxious-evoked baseline sensitivity in neonates to optimise analgesic trials

Maria M Cobo[1,2‡], Caroline Hartley[1‡], Deniz Gursul[1], Foteini Andritsou[1], Marianne van der Vaart[1], Gabriela Schmidt Mellado[1], Luke Baxter[1], Eugene P Duff[1,3], Miranda Buckle[1], Ria Evans Fry[1,4], Gabrielle Green[1], Amy Hoskin[1], Richard Rogers[5], Eleri Adams[4], Fiona Moultrie[1†], Rebeccah Slater[1†*]

[1]Department of Paediatrics, University of Oxford, Oxford, United Kingdom; [2]Universidad San Francisco de Quito USFQ, Colegio de Ciencias Biologicas y Ambientales, Quito, Ecuador; [3]Wellcome Centre for Integrative Neuroimaging, University of Oxford, Oxford, United Kingdom; [4]Newborn Care Unit, John Radcliffe Hospital, Oxford University Hospitals NHS Foundation Trust, Oxford, United Kingdom; [5]Department of Anaesthetics, John Radcliffe Hospital, Oxford University Hospitals NHS Foundation Trust, Oxford, United Kingdom

**Abstract** Despite the high burden of pain experienced by hospitalised neonates, there are few analgesics with proven efficacy. Testing analgesics in neonates is experimentally and ethically challenging and minimising the number of neonates required to demonstrate efficacy is essential. EEG (electroencephalography)-derived measures of noxious-evoked brain activity can be used to assess analgesic efficacy; however, as variability exists in neonate's responses to painful procedures, large sample sizes are often required. Here, we present an experimental paradigm to account for individual differences in noxious-evoked baseline sensitivity which can be used to improve the design of analgesic trials in neonates. The paradigm is developed and tested across four observational studies using clinical, experimental, and simulated data (92 neonates). We provide evidence of the efficacy of gentle brushing and paracetamol, substantiating the need for randomised controlled trials of these interventions. This work provides an important step towards safe, cost-effective clinical trials of analgesics in neonates.

*For correspondence:
rebeccah.slater@paediatrics.ox.ac.uk

†These authors contributed equally to this work
‡These authors also contributed equally to this work

Competing interests: The authors declare that no competing interests exist.

## Introduction

Considering the short-term stress and long-term neurodevelopmental impact associated with repeated pain exposure in early life (*Brummelte et al., 2012*; *Chau et al., 2019*; *Morison et al., 2001*; *Vinall et al., 2014*), effective pain relief is crucial in neonatal intensive care (*Hall and Anand, 2014*; *Lim and Godambe, 2017*). Nevertheless, as a result of the implicit challenges of measuring pain in neonates, and the ethical and experimental challenges of conducting neonatal clinical trials, few analgesics have proven efficacy in this population (*Allegaert, 2017*; *Moultrie et al., 2017*; *Slater et al., 2020*). Participants in clinical trials risk exposure to potential adverse effects, and therefore every effort should be made to minimise the sample size necessary to demonstrate efficacy (*European Medicines Agency, 2001*). However, as factors such as age (*Fabrizi et al., 2011*; *Green et al., 2019*; *Hartley et al., 2016*), prior pain experience (*Ozawa et al., 2011*; *Slater et al., 2010a*), stress (*Jones et al., 2017*), sex (*Bartocci et al., 2006*; *Verriotis et al., 2018*), illness (*Ranger et al., 2013*), and behavioural state (*Slater et al., 2006*) influence noxious-evoked responses, large sample sizes are often required to account for between-subject variability

**eLife digest** Hospitalized newborns often undergo medical procedures, like blood tests, without pain relief. This can cause the baby to experience short-term distress that may have negative consequences later in life. However, testing the effects of pain relief in newborns is challenging because, unlike adults, they cannot report how much pain they are experiencing.

One way to overcome this is to record the brain activity of newborns during a painful procedure and to see how these signals are modified following pain relief. Randomized controlled trials are the gold standard for these kinds of medical assessments, but require a high number of participants to account for individual differences in how babies respond to pain. Finding ways to reduce the size of pain control studies could lead to faster development of pain relief methods.

Here, Cobo, Hartley et al. demonstrate a way to reduce the number of newborns needed to test potential pain-relieving interventions. In the experiments, the brain activity of nine babies was measured after a gentle poke and after a painful clinically required procedure. Cobo, Hartley et al. found that the babies' response to the gentle poke correlated with their response to pain. Further data analysis revealed that this information can be used to predict the variability in pain experienced by different newborns, reducing the number of participants needed for pain relief trials.

Next, Cobo, Hartley et al. used this new approach in two pilot tests. One showed that gently stroking an infant's leg before blood is drawn from their heel reduced their brains' response to pain. The second showed that giving a baby the painkiller paracetamol lessened the brain's response to immunisation.

The new approach identified by Cobo, Hartley et al. may enable smaller studies that can more quickly identify ways to reduce pain in babies. Furthermore, this work suggests that gentle brushing and paracetamol could provide pain relief for newborns undergoing hospital acute procedures. However, more formal clinical trials are needed to test the effectiveness of these two strategies.

(*Anand et al., 2004*; *Ancora et al., 2013*; *Hartley et al., 2018*; *Kabataş et al., 2016*; *Sindhur et al., 2020*; *Taddio et al., 2006*).

In adult studies, cross-over trial designs are often used to minimise sample sizes by reducing between-subject variability (*Cooper et al., 2016*). However, this approach may not be appropriate when studying pain in neonates as painful medical procedures can only be performed when clinically necessary and within-subject variables that influence pain can change dramatically across sequential test occasions. One approach used to balance demographic characteristics or other prognostic factors across treatment groups in clinical trials is to stratify neonates across treatment arms and to adjust for these factors in the statistical analysis (*McEntegart, 2003*). While this can improve comparability across groups for recognised factors, many unknown variables likely influence pain sensitivity, and a more nuanced approach to account for individual differences in noxious-evoked sensitivity could be more effective in reducing sample sizes. In analgesic studies performed in adults, individual pain thresholds can be identified by applying graded increments of experimental stimulus intensity until pain is reported by the participants. This can be used to stratify treatment groups (*Demant et al., 2014*; *Smith et al., 2017*; *Vollert et al., 2017*) or statistically correct for variability in baseline pain thresholds (*Lane et al., 2010*; *Sanga et al., 2013*). In neonates, application of graded non-noxious stimuli such as von Frey hairs has previously been used to identify limb reflex withdrawal thresholds (*Andrews and Fitzgerald, 2002*; *Andrews and Fitzgerald, 1999*; *Andrews and Fitzgerald, 1994*; *Kühne et al., 2012*), but these have not been used as a baseline measure in analgesic clinical trials.

In the absence of a validated objective biomarker of pain (*Davis et al., 2020*), electroencephalography (EEG)-derived measures of brain activity may provide a valuable surrogate marker of pain by measuring the noxious-evoked activation of the cortex. In adults, brain activity during painful procedures is strongly correlated with verbal reports (*Coghill et al., 2017*); and in neonates, a template of noxious-evoked brain activity that discriminates between noxious and non-noxious procedures has been previously characterised and validated (*Hartley et al., 2017*). Here, we develop and test an experimental paradigm that assesses individual baseline sensitivity in neonates by measuring noxious-evoked brain activity in response to a low-intensity experimental noxious

stimulus, and demonstrate that accounting for this measure of noxious-evoked baseline sensitivity substantially reduces the sample sizes required in neonatal studies of analgesic efficacy. The term 'noxious-evoked baseline sensitivity' is used to refer to individual neonate's noxious-evoked baseline brain activity. This will be related to multiple neural and non-neural factors including nociceptive processing, arousal, attention, signal-to-noise ratio of the EEG recording, and differences in head size.

In contrast to studies in adults which have shown similar patterns of activity evoked by both painful and non-painful stimuli (*Mouraux et al., 2011*), we have previously shown that the pattern of brain activity that we analyse here is not evoked by visual, auditory, and tactile stimuli which evoke similar levels of physiological arousal (*Hartley et al., 2017*). In Study 1, we demonstrate that the magnitude of noxious-evoked brain activity in response to an experimental stimulus correlates with the magnitude of brain activity evoked by the clinical procedure and thus reflects baseline sensitivity. In Study 2, we use simulated data to demonstrate the increased statistical power that can be achieved by including baseline sensitivity as a covariate when analysing the effect of an intervention in small samples. In Study 3, we test this novel paradigm using a non-pharmacological pain-relieving intervention of known efficacy – gentle touch – prior to heel lancing. Finally, in Study 4 we investigate the analgesic efficacy of oral paracetamol given prior to immunisation in prematurely born neonates. Overall, we demonstrate that measuring and accounting for noxious-evoked baseline sensitivity could improve the design of analgesic efficacy investigations for this patient population.

## Results

### Study 1: Characterising individual noxious-evoked baseline sensitivity using brain activity in neonates

We hypothesised that a measure of neonatal noxious-evoked baseline sensitivity could be used to account for inter-individual variability in noxious-evoked brain activity in studies of analgesic efficacy (noxious-evoked sensitivity paradigm, *Figure 1*) and predicted that this would reduce the sample sizes needed in clinical trials. In order to use noxious-evoked brain activity in response to a mild experimental noxious stimulus as a measure of baseline sensitivity, it must be significantly and strongly correlated with the response to the clinically required procedure. In Study 1, we therefore assessed the feasibility and initial validity of the paradigm by investigating the relationship between individual responses of term neonates to experimental noxious stimuli and a subsequent clinically required heel lance. This was a retrospective study presenting previously unpublished data from term neonates studied between 2014 and 2015 who had received both a heel lance and experimental noxious stimuli on the same test occasion (n = 9). Whilst this sample size is small, this was a feasibility study and this relationship is retested in Study 3. The magnitudes of the noxious-evoked brain activity evoked by stimulating a neonate's foot with a controlled mild experimental noxious stimulus (force = 64 mN; magnitude range 0.15–0.62, *Figure 2A*) were strongly correlated with the magnitude of the noxious-evoked brain activity generated by a clinically required heel lance (range −0.07 to 2.34) in the same neonates (p=0.0025, $R^2 = 0.77$, *Figure 2A*). Therefore, application of a mild experimental noxious stimulus prior to performing a clinically required painful procedure could provide a novel measure of neonatal baseline sensitivity, which could be used as a covariate in studies of analgesic efficacy to account for inter-individual variability in pain responses (*Figure 1*). In the following sections, we simulate and test the impact of applying this novel paradigm in studies investigating the efficacy of pain-relieving interventions.

### Study 2: Simulating the effect of accounting for individual baseline sensitivity

In Study 2, we used simulated data to investigate whether accounting for individual differences in noxious-evoked baseline sensitivity has the potential to reduce the sample size needed to assess the efficacy of an analgesic intervention. Here, we initially assume that an effective analgesic intervention results in a 40% reduction in noxious-evoked brain activity; this is clinically meaningful as a similar reduction in noxious-evoked brain activity is observed when adults report significantly lower verbal pain scores (*Lorenz et al., 1997*; *von Mohr et al., 2018*). We simulated an Intervention Group and

**Figure 1.** Noxious-evoked sensitivity paradigm explained. Schematic representation of the noxious-evoked sensitivity paradigm components. A brief description of each step is included with additional explanatory notes.

Control Group across a range of sample sizes, simulating both baseline sensitivity data and responses to heel lance, and assuming the relationship between these measures observed in Study 1.

The simulated Control Group and the Intervention Group responses were compared using a linear regression with and without baseline sensitivity as a covariate (see 'Materials and methods'). At a significance level of 0.05, the sample size to achieve a given power is substantially reduced when baseline sensitivity is accounted for (*Figure 2B*). The reduction in sample size that can be achieved by accounting for individual differences in baseline sensitivity is highly dependent on the anticipated effect size of the intervention (*Figure 2C,D*). For example, at the extremes we considered, with an assumed intervention effect size of 95% (and power of 95%), a sample of 11 neonates per group would be required without accounting for baseline sensitivity compared with eight neonates per group when baseline sensitivity is accounted for, a 27% reduction in sample size. Whereas, by comparison, assuming an intervention effect size of 5%, the sample size required to achieve 95% power is 5458 neonates per group without accounting for individual baseline sensitivity, compared with 660 neonates per group when baseline sensitivity is accounted for – representing an 88% reduction in sample size (*Figure 2C,D*). Assuming an intervention effect size of 40%, a sample size of 16 neonates per group (32 neonates in total) would be sufficient to observe a significant intervention effect with 95% power if individual differences in baseline sensitivity are accounted for. In contrast, a sample size of 66 neonates per group (132 neonates in total) is required to achieve the same power if neonatal baseline sensitivity is not accounted for (*Figure 2B*).

The percentage reduction in sample size which can be achieved by accounting for individual differences in baseline sensitivity is also dependent on the strength of the correlation between the brain responses evoked by the experimental stimuli and the acute clinical procedure (*Figure 2E*). When a weak correlation exists between the measures (calculated using the standard deviation of

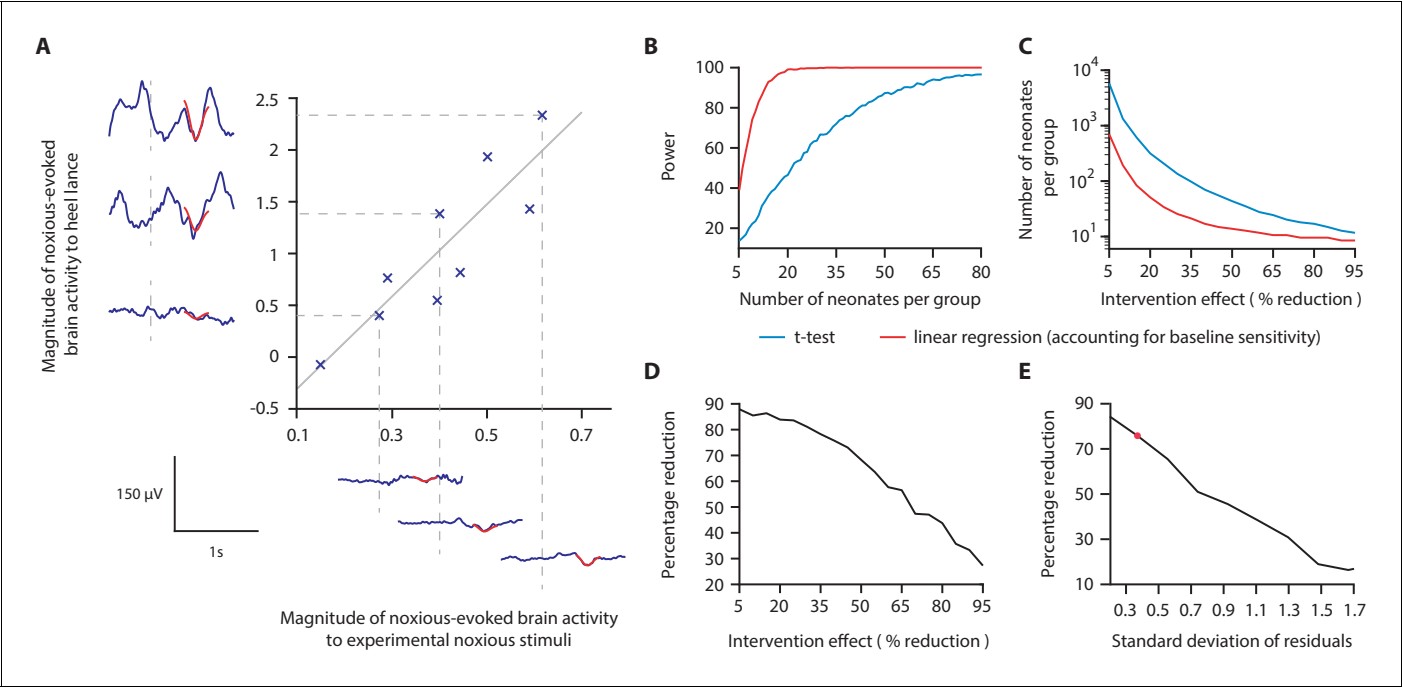

**Figure 2.** Magnitude of noxious-evoked brain activity in response to experimental noxious stimuli correlates with the response to a clinically required heel lance and can be used as a measure of baseline sensitivity to reduce sample sizes. (A) The magnitude of noxious-evoked brain activity following mild experimental noxious stimuli and a clinically required heel lance was significantly correlated within-subject (p=0.0025, $R^2$ = 0.77, n = 9, Pearson correlation test, Study 1); grey solid line indicates line of best fit. Dashed lines and their corresponding electroencephalography (EEG) traces indicate three neonates with a range of response magnitudes. The magnitude of the brain activity was quantified using a template of noxious-evoked activity, shown overlaid in red (*Hartley et al., 2017*). (B-E) In Study 2, we used simulated data to investigate how sample size is altered when the relationship in (A) is considered. (B) For each sample size, 1000 data sets were simulated with a 40% reduction in the response to a clinically required procedure assumed in the Intervention Group. The power (percentage of significant results, p<0.05) to detect a difference between the two groups was calculated for each sample size using a linear regression with (red) and without (blue) accounting for individual differences in baseline sensitivity. (C) The number of neonates required to achieve 95% power with different levels of intervention effect. Simulations were run with increasing numbers of neonates until 95% power was achieved. (D) Percentage reduction in the number of neonates required per group when individual baseline sensitivity is accounted for compared with not accounting for baseline sensitivity (power = 95%). (E) The percentage reduction in the number of neonates required per group with different degrees of correlation (standard deviation of residuals) between the responses to experimental noxious stimuli and clinically required procedure (40% intervention effect, 95% power). The red marker indicates the standard deviation of residuals (SD = 0.37) in (A) (*Figure 2—source data 1*). The code to produce (B-E) is available from https://gitlab.com/paediatric_neuroimaging/simulating_power_nociceptive_sensitivity.git.
The online version of this article includes the following source data for figure 2:

**Source data 1.** Numerical data plotted in *Figure 2A*.

the correlation residuals), the reduction in sample size is low. Conversely, with a strong correlation between measures, a greater reduction in sample size is achieved. For example, with an assumed intervention effect of 40% and the low standard deviation of the residuals observed in Study 1 (SD of residuals = 0.37), accounting for baseline sensitivity results in a sample size reduction of approximately 76%, compared to a sample size reduction of 17% when a high noise level (SD of residuals = 1.7) is observed in the correlation (*Figure 2E*).

## Study 3: Testing the paradigm: a non-pharmacological pain-relieving intervention study

In a previous study, we reported that a non-pharmacological gentle touch intervention (brushing a neonate's leg at a rate of approximately 3 cm/s to optimally stimulate C-tactile fibres) prior to a clinically required heel lance caused a 40% reduction in noxious-evoked brain activity (*Gursul et al., 2018*). In Study 3, we used the same non-pharmacological intervention in an independent prospective cohort of healthy neonates that clinically required a heel lance for the purpose of blood sampling and tested the effect of incorporating the noxious-evoked baseline sensitivity paradigm and accounting for inter-individual differences in baseline sensitivity. Based on power calculations from

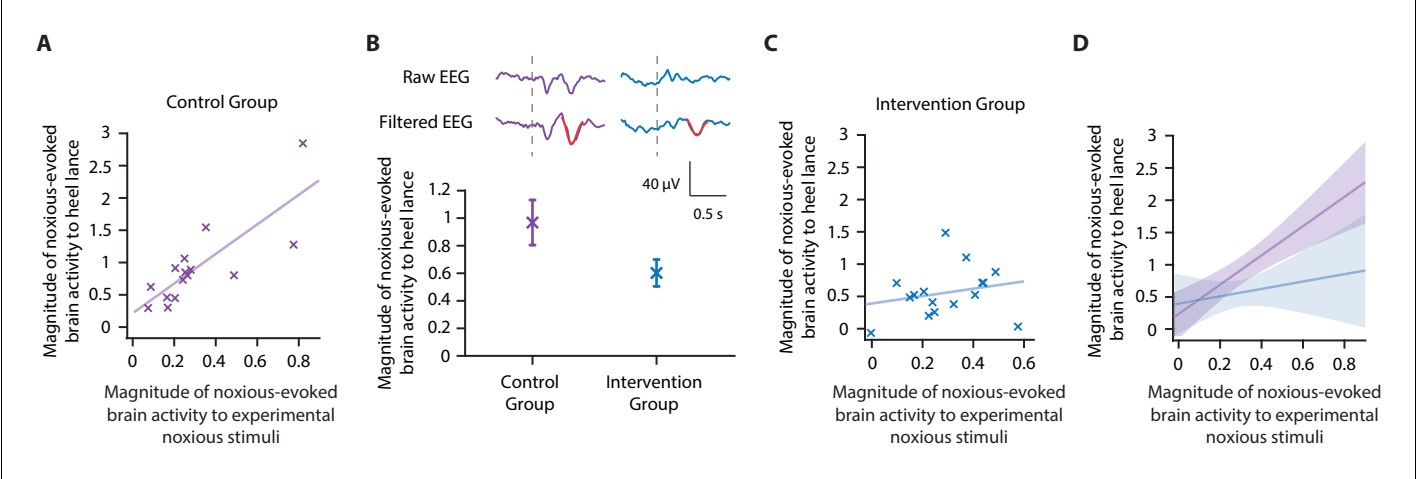

**Figure 3.** Accounting for individual baseline sensitivity in the assessment of efficacy of a gentle touch intervention. (A) The magnitude of the noxious-evoked brain activity following a mild experimental noxious stimulus compared with the clinically required heel lance for each neonate in the Control Group (n = 15). Solid line indicates line of best fit. (B) (Top) Group average raw electroencephalography (EEG) and (Woody) filtered EEG traces in response to the clinically required heel lance; Control Group (purple) and Intervention Group (neonates received gentle touch at a rate of approximately 3 cm/s for 10 s prior to the heel lance, n = 16) (blue). Dashed lines indicate the point of stimulation; the template of noxious-evoked brain activity is shown overlaid in red. Each neonate's EEG responses to the experimental noxious stimulus and the heel lance are shown in *Figure 3—figure supplement 1*. (Bottom) Magnitude of the noxious-evoked brain activity following heel lance in the two groups. Error bars indicate mean ± standard error. (C) Comparison of the stimulus responses for each neonate in the Intervention Group. Gentle touch was not applied prior to the experimental noxious stimuli so that each neonate's individual baseline sensitivity could be assessed. (D) Confidence intervals of the correlations for the two groups shown overlaid: Control Group (purple), Intervention Group (blue), solid lines indicate line of best fit. The effect of the intervention (gentle touch) is demonstrated by the difference between the two groups' confidence intervals and is most evident in neonates who have greater baseline sensitivity (i.e. higher responses to the experimental noxious stimulus) (*Figure 3—source data 1*).

The online version of this article includes the following source data and figure supplement(s) for figure 3:

**Source data 1.** Numerical data plotted in *Figure 3A,B,C*.

**Figure supplement 1.** Noxious-evoked brain activity in individual neonates in the Control Group and Intervention Group, Study 3.

simulated data in Study 2, assuming a 40% reduction in noxious-evoked brain activity from the intervention and 95% power, a total of 16 neonates were included in the Intervention Group and were gently brushed on the leg ipsilateral to the stimulus site at a rate of approximately 3 cm/s for 10 s prior to heel lancing (*Gursul et al., 2018*). A further 15 neonates were included in the Control Group where the heel lance was performed without gentle brushing. All neonates received mild experimental noxious stimulation prior to heel lancing to assess their individual baseline sensitivity (see 'Materials and methods'). Unlike Study 1, in which neonates had been stimulated with a force of 64 mN, a force of 128 mN was applied in this prospective cohort to increase the signal-to-noise ratio. The necessary strong correlation between the evoked response to the experimental stimulus and clinical procedure was confirmed in the Control Group (p=0.0013, $R^2$ = 0.65, *Figure 3A*).

Consistent with the previously published study (*Gursul et al., 2018*), the gentle touch intervention resulted in a 39% reduction in the magnitude of the noxious-evoked brain activity, but a significant intervention effect was not observed (although the result indicated borderline significance) when baseline sensitivity was not accounted for, likely due to the lack of power with this sample size (linear regression, t = 1.95, p=0.05, *Figure 3B*, Study 2 indicates a power of 40% for a sample of this size without accounting for baseline sensitivity, *Figure 2B*). However, when noxious-evoked baseline sensitivity was accounted for as a covariate in the analysis, a significant intervention effect was observed (linear regression, t = 2.29, p=0.026).

To further understand these results, we compared the relationship between the responses within the Control Group and Intervention Group. Unlike the significant correlation between the magnitude of noxious-evoked brain activity in response to the experimental noxious stimuli and heel lancing demonstrated in the Control Group (p=0.0013, $R^2$ = 0.65, *Figure 3A*), this relationship was disrupted in the Intervention Group (p=0.39, $R^2$ = 0.05, *Figure 3C*). In particular, we observed reduced

noxious-evoked brain activity following the gentle brushing intervention in neonates with high baseline sensitivity (*Figure 3D*), suggesting that the effect of pain-relieving interventions is most prominent in neonates with greater noxious-evoked baseline sensitivity.

## Testing the paradigm with other modalities: noxious-evoked reflex withdrawal

Noxious stimulation in neonates evokes a range of physiological responses including facial grimacing, reflex withdrawal, and physiological responses (*Cornelissen et al., 2013*; *Hartley et al., 2015*; *Hatfield and Ely, 2015*). There is great value in establishing whether accounting for individual differences in baseline sensitivity can be applied to other pain-related measures. In Study 3, the magnitude of the reflex withdrawal was also recorded in response to the experimental noxious stimulation and heel lancing. In the Control Group, the magnitude of the reflex withdrawal response to experimental noxious stimulation was significantly correlated with the reflex withdrawal evoked by heel lancing (p=0.009, $R^2$ = 0.36, *Figure 4A*). However, this correlation in reflex withdrawal activity was weaker than the relationship in the noxious-evoked brain activity, limiting its use (*Figure 2E*). Assuming an intervention effect of 40% and this level of correlation identified within the same size sample, simulated data reveals that accounting for baseline sensitivity using noxious-evoked reflex activity provides only 17.3% power to detect a significant difference between the two groups compared with a power of 11.3% without accounting for baseline sensitivity.

In this study, the gentle touch intervention did not significantly reduce the magnitude of the reflex withdrawal activity following heel lancing, either when accounting for baseline sensitivity (linear regression, t = −1.43, p=0.17) or without accounting for baseline sensitivity (t = −1.73, p=0.10, *Figure 4B*). While it is possible that reflex withdrawal of the stimulated limb is not modulated by gentle touch, as has previously been suggested (*Gursul et al., 2018*), the intervention clearly disrupted the correlation between baseline reflex sensitivity and the reflex evoked by heel lancing

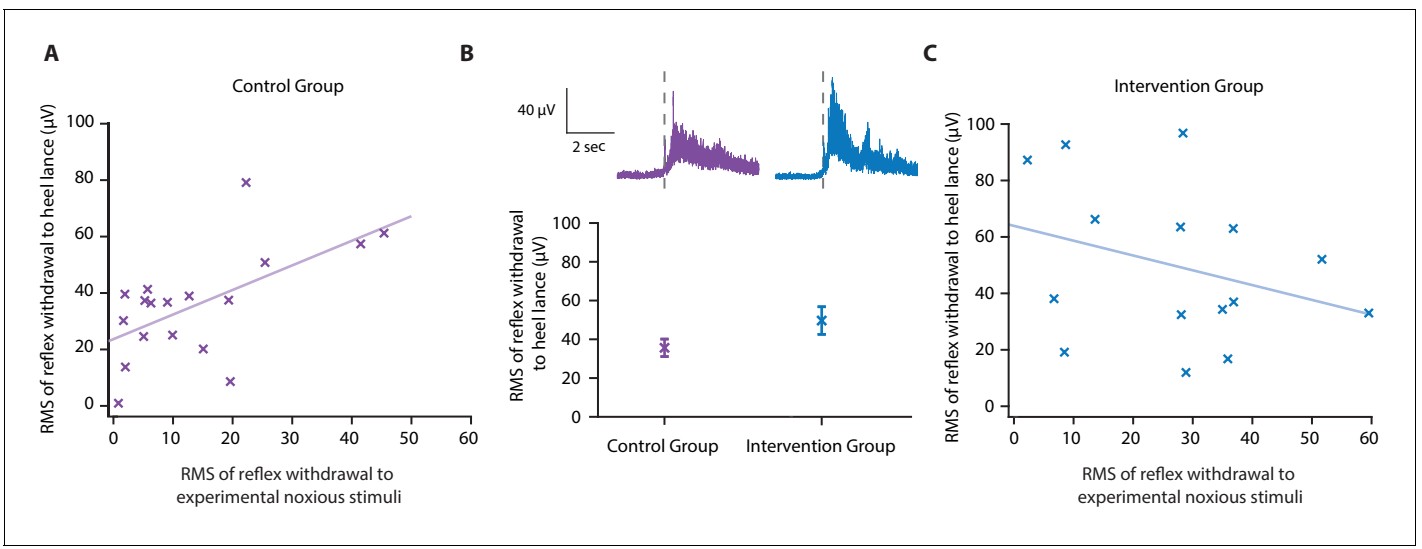

**Figure 4.** Effect of gentle touch on reflex responses. (**A**) The magnitude of the reflex withdrawal following a mild experimental noxious stimulus (baseline reflex sensitivity) compared with the clinically required heel lance for each neonate in the Control Group (n = 18). Solid line indicates line of best fit. (**B**) Average electromyography (EMG) traces (top) for neonates in the Control Group (purple) and Intervention Group (blue, n = 15) where neonates were gently brushed at a rate of approximately 3 cm/s for 10 s prior to the heel lance. Dashed lines indicate the point of stimulation. Each neonate's EMG responses to the experimental noxious stimulus and the heel lance are shown in *Figure 4—figure supplement 1*. (Bottom) Magnitude of the reflex withdrawal response in the two groups. Error bars indicate mean of the root mean square (RMS) of the reflex withdrawal ± standard error. (**C**) The magnitude of the reflex withdrawal following a mild experimental noxious stimulus (baseline reflex sensitivity) compared with the clinically required heel lance for each neonate in the Intervention Group (gentle touch) (*Figure 4—source data 1*).

The online version of this article includes the following source data and figure supplement(s) for figure 4:

**Source data 1.** Numerical data plotted in *Figure 4A,B,C*.

**Figure supplement 1.** Reflex withdrawal activity in individual neonates in the Control group and Intervention Group, Study 3.

(p=0.25, $R^2$ = 0.1, *Figure 4C*). The brushing intervention may have caused a change in baseline muscle activity in some individuals resulting in the larger residuals in the Intervention Group (SD of the residuals – Control Group 15.3 µV; Intervention Group 26.2 µV).

## Study 4: A pharmacological analgesic study

In Study 4, we conducted an opportunistic study to investigate whether the administration of paracetamol prior to immunisation significantly reduces noxious-evoked brain activity. In 2015, national clinical guidelines recommended the administration of paracetamol at the time of meningitis B immunisation due to its antipyretic effect (*NHS England and Public Health England, 2015*). Therefore, our local neonatal unit (John Radcliffe Hospital) began administering oral paracetamol to neonates immediately after vaccination. In October 2018, the local practice guidelines were updated, recommending the administration of oral paracetamol 1 hr pre-vaccination. Prior to the guideline change, we studied 16 neonates who did not receive paracetamol before immunisations (Control Group), recording their noxious-evoked brain activity during immunisations. Following the guideline change, we recorded noxious-evoked brain activity in 16 neonates who received paracetamol 1 hr prior to immunisations (Intervention Group) (see 'Materials and methods' and *Figure 5A*). In the Intervention Group, we explored the relationship between noxious-evoked baseline sensitivity and brain activity evoked by immunisation following paracetamol administration.

Noxious-evoked brain activity in response to immunisation was characterised using a fast frame rate video camera to identify the time when the needle first came into contact with the skin (*Hartley et al., 2014*; *Verriotis et al., 2016*). For each neonate, up to three immunisations were recorded on the same test occasion. First, we validated the use of the template of noxious-evoked brain activity (*Hartley et al., 2017*) to quantify the magnitude of noxious-evoked brain activity from immunisation applied to the thigh (see Methods to validate the template of noxious-evoked brain activity, *Figure 5—figure supplement 1* and *Figure 5—figure supplement 2*). The magnitude of noxious-evoked brain activity following immunisation was significantly lower in the neonates who received paracetamol prior to vaccination (linear mixed effects regression model with subject and number of immunisations set as random effects, Control Group mean 0.88 [SD 0.58] (n = 15); Intervention Group mean 0.40 [SD 0.30] (n = 14), t = 3.61, p<0.001, *Figure 5B*).

In a subset of 11 of the 16 neonates in the Intervention Group, who received paracetamol prior to immunisation, we also recorded responses to experimental noxious stimulation before and approximately 1 hr after paracetamol administration (Intervention Group subset, *Figure 5A*). As this study was implemented opportunistically following changes in clinical guidelines, responses to experimental noxious stimuli were not recorded in all neonates. Nevertheless, the baseline sensitivity measures that were recorded in response to the experimental noxious stimuli applied prior to paracetamol administration had a range of values that were similar to Studies 1 and 3 (range: 0.09–0.77). Likewise, the magnitude of the brain activity evoked by the immunisations was similar to that evoked by heel lance in the previous studies for both the Intervention Group (Intervention Group [Study 3] range −0.06 to 1.48; Intervention Group [Study 4] range −0.08 to 1.22) and the Control Group (Control Group [Study 3] range 0.30–2.84; Control Group [Study 4] range −0.07 to 2.14). Although we did not record the baseline sensitivity in neonates in the Control Group, in the absence of a pain-relieving intervention, we would expect the response to be correlated with noxious-evoked brain activity evoked by immunisation. As the correlation between baseline sensitivity and response to immunisation was low in the Intervention Group (p=0.12, $R^2$ = 0.33, n = 9, *Figure 5C*), the relationship between these measures was likely disrupted by paracetamol. Similar to the gentle brush intervention, the neonates with high baseline sensitivity, represented by a high magnitude response to experimental noxious stimulation prior to paracetamol administration, had much lower magnitude responses to immunisation than would have been expected without an analgesic intervention (*Figure 5C*). Similarly, the correlation between the baseline sensitivity (magnitude of the noxious-evoked baseline sensitivity prior to paracetamol administration) and the response to the experimental noxious stimuli 1 hr post-paracetamol administration was disrupted (p=0.83, $R^2$ = 0.006, n = 9, *Figure 5D*). There was no significant difference in the responses to the experimental noxious stimuli before and after paracetamol administration (linear regression, before paracetamol mean: 0.27 [SD 0.38]; after paracetamol mean: 0.27 [SD 0.35], t = 0.17, p=0.86, n = 9) but we were likely not powered to observe an effect.

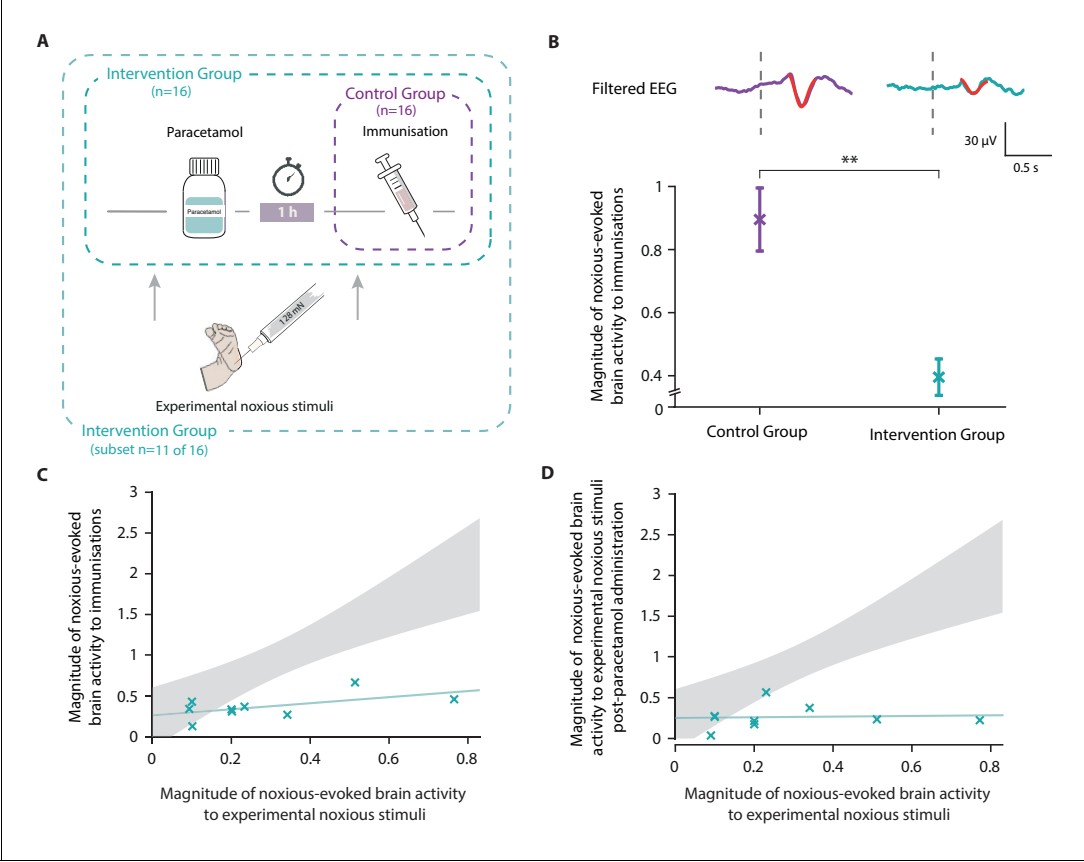

**Figure 5.** Paracetamol significantly reduces noxious-evoked brain activity following immunisation. (**A**) Experimental design of Study 4. Electroencephalography (EEG) was recorded during immunisations in neonates before (Control Group, n = 16) and after the guideline change (Intervention Group, n = 16, received paracetamol 1 hr prior to immunisation). A subset of neonates in the Intervention Group (n = 11 of 16) received experimental noxious stimuli before and approximately 1 hr after paracetamol administration. (**B**) Average (Woody) filtered EEG following immunisations are displayed (top); for the Control Group in purple and Intervention Group in teal, the template of noxious-evoked brain activity is shown overlaid in red. Dashed lines indicate the point of contact of the needle with the skin. The use of the template of noxious-evoked brain activity to quantify the magnitude of noxious-evoked brain activity from immunisation applied to the thigh was validated: *Figure 5—figure supplement 1* and *Figure 5—figure supplement 2*. Individual neonate's EEG responses to the immunisation are shown in *Figure 5—figure supplement 3*. (Bottom) Magnitude of the noxious-evoked brain activity following immunisations in the two groups (Control Group n = 15, Intervention Group n = 14), error bars indicate mean ± standard error (linear mixed effects regression model − without accounting for baseline sensitivity, t = 3.61, **p<0.001). (**C**) Magnitude of the noxious-evoked brain activity following the experimental noxious stimulus prior to paracetamol administration (baseline sensitivity) compared with the noxious-evoked brain activity to immunisation (which was approximately 1 hr after paracetamol administration) for each neonate in the Intervention Group subset (n = 9, markers in teal). For comparison, the confidence interval of the Control Group correlation in Study 3 (i.e. the correlation between the response to experimental noxious stimuli and a heel lance) is shown in grey. (**D**) Magnitude of the noxious-evoked brain activity to experimental noxious stimuli prior to paracetamol administration (baseline sensitivity) compared with the noxious-evoked brain activity to experimental noxious stimuli applied approximately 1 hr after paracetamol administration for each neonate in the Intervention Group subset (n = 9, markers in teal). For comparison, the confidence interval of the Control Group correlation in Study 3 (i.e. the correlation between the response to experimental noxious stimuli and a heel lance) is shown in grey (*Figure 5—source data 1*).

The online version of this article includes the following source data and figure supplement(s) for figure 5:

**Source data 1.** Numerical data plotted in *Figure 5B,C,D*.

**Figure supplement 1.** Latency of the noxious-evoked brain activity in response to stimulation on the foot, thigh, and hand.

**Figure supplement 2.** Validation of the template for use in immunisation studies.

**Figure supplement 3.** Noxious-evoked brain activity following immunisations in individual neonates in the Control Group and Intervention Group, Study 4.

## Discussion

We demonstrate that accounting for individual differences in noxious-evoked baseline sensitivity significantly reduces the sample size required to assess the efficacy of analgesics in neonates. Noxious-evoked brain activity in response to a low-intensity experimental noxious stimulus can be used in neonates as a marker of baseline sensitivity and is highly correlated with the magnitude of noxious-evoked brain activity produced by clinically required acute painful procedures. Using both simulated and experimental data, we demonstrate that the sample size required to observe the effects of analgesic interventions (for a given power and significance level) can be significantly reduced when noxious-evoked baseline sensitivity is accounted for. Importantly, the percentage reduction in sample size is related to the expected effect size of the intervention and the degree of correlation between the baseline sensitivity measure and the brain activity evoked by the clinical procedure. By testing this novel paradigm in clinical studies, we re-confirm the efficacy of gentle touch as a non-pharmacological intervention that reduces brain activity evoked by heel lancing (*Gursul et al., 2018*) and we provide evidence to suggest that oral paracetamol is a candidate analgesic drug for procedural pain in neonates. Although these studies have a number of limitations (including lack of randomisation) and only investigate one aspect of the neonatal response to noxious input (namely an EEG-derived noxious-evoked potential), they provide strong evidence to suggest that randomised clinical trials investigating the efficacy of both gentle touch and paracetamol through multi-modal pain assessment measures are warranted.

Minimisation of sample sizes is imperative in clinical research, and particularly in neonatal studies given the inherent ethical, recruitment, and experimental challenges associated with studying this patient population. Considering that inter-individual variability drives increases in sample sizes required to demonstrate efficacy, addressing baseline variability is key. The paradigm we present here likely accounts for multiple factors affecting noxious-evoked baseline sensitivity in neonates including potential effects from prior pain exposure during hospitalisation (*Grunau et al., 2001*; *Johnston and Stevens, 1996*) and prematurity (*Slater et al., 2010a*). This provides a robust approach to indirectly control for a vast array of known and unknown demographic and environmental factors that influence noxious-evoked brain activity and result in inter-individual variability in responses, as well as potential experimental confounds which differ between individuals (such as differences in signal-to-noise ratios, head circumference, and skull thickness). Responses to other modalities such as visual, auditory, or tactile stimuli could be used to obtain a measure of baseline sensitivity, and background resting state brain activity is also predictive of individual noxious-evoked responses (*Baxter et al., 2021*). However, the aim of this study was to develop an experimental paradigm that accounts for the maximum variability in responses to acute painful procedures, to maximise the power to detect a true effect of an intervention. Applying an experimental noxious stimulus to obtain a measure of baseline sensitivity optimises the model as it optimally matches the main characteristics of a response that would be evoked by a procedure of clinical interest (e.g. the stimulus can be applied to the same body location, evokes noxious activity as well as other sensory-related brain activity, and it is measured at the same electrode site). Given the study aim was to reduce sample size in studies investigating acute pain, it is most appropriate to adjust for inter-individual variability using measures of noxious-evoked baseline sensitivity. In contrast, if alternative studies considered other sensory modalities, for example, visual processing, then a better measure of inter-individual baseline variability would be achieved using a visual stimulus.

The experimental noxious stimulus used in these studies provides a practical and ethical paradigm for the assessment of baseline sensitivity in neonates. It is non-tissue damaging in both term and ex-premature neonates, activates A∂ and C fibres (*van den Broeke et al., 2015*), does not evoke changes in facial expression or signs of behavioural distress (*Goksan et al., 2015*; *Hartley et al., 2017*; *Hartley et al., 2015*), and is acceptable to parents. The application of experimental noxious stimuli provides a reliable measure of baseline sensitivity as the mild stimuli can be repeated and trial averages calculated within individual neonates; an approach that substantially reduces the signal-to-noise ratio as compared with responses recorded in response to a single clinical procedure. Moreover, there is no evidence from our data that the experimental noxious stimuli increase the magnitude of the heel lance response given that the responses to heel lance reported here are similar to previous papers where the experimental noxious stimuli were not applied (*Hartley et al., 2017*), suggesting it is appropriate for use in a clinical setting. Despite the advantage of using this

approach, we cannot rule out the potential effects of selection bias (*Bishop, 2020*). A relatively high number of trials were rejected due to artefacts, which may be more pronounced when there are stimulus-related movements. If these movements are indicative of a more vigorous response to the noxious input, then it is plausible that we are unavoidably biased towards a subset of the population.

The applicability of the noxious-evoked baseline sensitivity paradigm was tested in the context of a pain-relieving intervention that we have previously shown to be effective in reducing noxious-evoked brain activity – gentle touch (*Gursul et al., 2018*). Neonates were gently brushed at a speed of 3 cm/s, which is approximately equivalent to the rate at which parents will naturally stroke their neonates (*Croy et al., 2016*) and optimises stimulation of C-tactile fibres (*Löken et al., 2009*). In an independent population of neonates, we re-confirmed that brushing the skin prior to a clinically required heel lance significantly reduces noxious-evoked brain activity. We used our noxious-evoked baseline sensitivity paradigm to indirectly account for many factors that influence the magnitude of noxious-evoked brain activity. In addition, we did not observe a significant difference in reflex withdrawal activity between the control neonates and the neonates who received gentle touch prior to the heel lance, which is consistent with our previous observations (*Gursul et al., 2018*). It is possible that either the magnitude of the reflex withdrawal is genuinely not modulated by the brush intervention or that a modulation in reflex activity would only be observed with a larger sample size. Importantly, a significant but weak correlation was observed between the reflex activity in response to the noxious stimuli and in response to heel lancing in the Control Group, suggesting that the paradigm presented here could be useful in future trials where reflex withdrawal activity is used as an outcome measure. As pain perception is a highly complex sensory and subjective emotional experience generated in the brain (*IASP, 2020*), quantifying noxious-evoked brain activity may represent a better proxy pain measure, and a more sensitive marker of analgesic efficacy, compared with reflex signals generated by the spinal cord.

In addition to minimising sample sizes, assessing baseline sensitivity may also allow for identification of neonates that would benefit most from analgesic interventions. Neonates with larger noxious-evoked baseline sensitivity had the greatest reduction in response following the intervention. In contrast, neonates with low baseline sensitivity were less likely to demonstrate a benefit of the intervention, as for this clinical procedure the potential reduction in their responses was minimal. This could be due to a floor effect whereby for some neonates noxious-evoked brain responses to heel lance is close to zero and cannot be reduced further. Improving our understanding of inter-individual variability in pain-related responses is pivotal to ensure that for each individual neonate potential adverse effects of analgesics are carefully weighed against potential benefits.

In our final study, we demonstrate that paracetamol significantly reduced the magnitude of the noxious-evoked brain activity following immunisation compared with neonates who did not receive paracetamol prior to immunisation. Although this result is consistent with studies in adults, using both EEG (*Bromm et al., 1992*; *Pickering et al., 2013*) and fMRI (*Pickering et al., 2015*), where paracetamol has been shown to reduce brain activity evoked by noxious procedures, previous studies in neonates have provided insufficient evidence to determine the analgesic efficacy of paracetamol for acute procedural pain (*Ohlsson et al., 2020*). While several studies suggest an opioid-sparing effect of paracetamol (*Ceelie et al., 2013*; *Härmä et al., 2016*) and reduced need for pain relieveing interventions (*Höck et al., 2020*), the majority of studies do not demonstrate a reduction in behavioural or physiological responses to commonly performed painful procedures, such as heel lancing (*Badiee and Torcan, 2009*; *Bonetto et al., 2008*; *Shah et al., 1998*) and invasive eye examinations to screen for retinopathy of prematurity (*Seifi et al., 2013*). The behavioural outcome measures used in these studies may fail to discriminate between pain and distress (*Moultrie et al., 2017*; *Slater, 2019*), which could limit conclusions related to analgesic efficacy. However, given the small sample size of the present study and that we are only characterising the immediate noxious-evoked brain activity following the needle insertion, rather than the activity associated with the injection of the fluid into the muscle for example, randomised clinical trials that also include other pain-related measures are warranted to assess the benefit of paracetamol administration prior to immunisation. Nonetheless, small studies in adults demonstrate that candidate drugs can modulate pain-related neural activity in the absence of verbally reported analgesia, and these brain-derived measures are recognised as a valuable approach to obtain objective evidence related to potential analgesic efficacy in early proof of concept studies (*Wanigasekera et al., 2018*). The noxious-evoked brain

activity measure used here quantifies the evoked potential produced at the central vertex electrode site (Cz), which has been shown to have the greatest and most reproducible response size amplitude compared to other electrodes sites across the brain (*Hartley et al., 2017*; *Verriotis et al., 2016*). This measure does not represent all nociceptive activity across the brain and cannot be used to investigate the various aspects of pain perception (*Hartley et al., 2017*); a multi-modal approach to pain assessment is therefore important in follow-on studies (*van der Vaart et al., 2019*). However, in the absence of verbalisation, neuroimaging methods provide an objective proxy approach which has been used to infer pain perception following noxious events (*Baxter et al., 2021*; *Duff et al., 2020*; *Gursul et al., 2019*; *Hartley et al., 2017*).

Paracetamol is administered as an antipyretic for the meningitis B immunisations. An update to our local clinical guidelines was implemented, whereby the paracetamol was administered prior to rather than after immunisation. This meant we were able to opportunistically study whether paracetamol can reduce noxious-evoked brain activity following immunisation. Our study is significantly limited due to the restricted sample sizes, lack of randomisation and blinding, and because in the Control Group, where paracetamol was administered after immunisation, we did not record baseline sensitivity prior to immunisation. Therefore, we do not have data to confirm that the baseline sensitivity is correlated with the magnitude of the evoked brain activity following immunisation; although, given there is no discernible correlation between these measures in the Intervention Group, this strongly suggests that this relationship has been disrupted by paracetamol administration. Furthermore, for neonates with high baseline sensitivity, the brain responses evoked by the immunisation were much lower than would be expected in the absence of the analgesic intervention. To broaden the utility of this paradigm, it will be necessary to characterise the correlation between baseline sensitivity and a range of acute clinical procedures, including immunisation.

Although many factors that influence individual variability in responses are accounted for using our noxious-evoked sensitivity paradigm, it does not account for differences in rapidly fluctuating state effects such as differences in attention or sleep state that could vary between the baseline sensitivity testing and the implementation of the clinical procedure. Understanding how state differences influence variability in noxious-evoked responses will facilitate better estimation of the expected responses to clinically required painful procedures. A recent fMRI study demonstrated that noxious-evoked brain activity can be predicted from a neonate's resting state brain activity as well as the structural integrity of key white matter pathways (*Baxter et al., 2021*). Investigating the role of baseline EEG activity and exploring the neurological differences underlying variability in the noxious-evoked brain activity described here could further improve the utility of the paradigm. In addition, while our paradigm is applicable to many of the most common acute somatic painful procedures which neonates are exposed to including heel lancing, cannulation, and injections, this paradigm may not apply to many types of pain such as visceral pain, post-operative pain, longer procedures, such as retinopathy of prematurity screening, procedures with a slow-rising onset, or chronic pain.

In summary, the assessment of pain in non-verbal neonates is challenging (*Slater, 2019*) and the wide variability in individual responses to painful procedures complicates the assessment of analgesics. Currently there is a paucity of evidence regarding the efficacy of pain-relieving interventions used in neonatal practice (*Baarslag et al., 2017*). Here, we present a paradigm that accounts for individual noxious-evoked baseline sensitivity and we demonstrate its utility in terms of sample size reduction. Using this paradigm in clinical trials could optimise resources, maximise the value of collected data, and ultimately expedite the discovery and validation of urgently needed analgesics for this patient population.

## Materials and methods

### Study design and participants

A total of 92 neonates were included in four studies. In Study 1, the relationship between responses to experimental noxious stimuli and clinically required heel lance was investigated in nine neonates using unpublished data previously collected for other studies. In Study 2, the potential value of the statistical relationship identified in Study 1 was investigated using computational simulations. In Study 3, brain activity and reflex withdrawal responses from 38 neonates were recorded to test the paradigm with gentle touch as a pain-relief intervention. In Study 4, brain activity was recorded from

29 neonates in response to immunisations to test the analgesic efficacy of paracetamol. Additionally, the brain-derived measures to characterise immunisation-evoked activity were validated in a further 16 neonates.

The participants were recruited from the Maternity Unit and Newborn Care Unit at the John Radcliffe Hospital, Oxford University Hospitals National Health Service Foundation Trust, Oxford, UK. Medical charts were reviewed, and neonates assessed as clinically stable, not receiving analgesics at the time of study (except from paracetamol where specified), and with no history of neurological problems or maternal substance abuse were eligible for inclusion. Participant demographics are presented in *Table 1*. The estimate of cumulative prior pain exposure was quantified from each neonate's clinical records as the total number of acute skin-breaking procedures (including heel lances, venepuncture, and intravenous cannulations) and aspirations (oropharyngeal or endotracheal) from time of birth to time of study (*Hartley et al., 2016*). These procedures were chosen based on a prospective epidemiology study describing the most commonly performed clinical procedures neonates are exposed to during hospitalisation (*Carbajal et al., 2008*).

## Research governance

Studies were conducted in accordance with the Declaration of Helsinki and Good Clinical Practice guidelines. Ethical approval was obtained from the National Research Ethics Service (reference 12/SC/0447) and informed written parental consent was obtained prior to each study.

## Experimental design

### Study 1: Characterising individual noxious-evoked baseline sensitivity using brain activity in neonates

The aim of this study was to investigate the relationship between noxious-evoked brain activity in response to experimental stimuli and clinically required heel lance within-subjects in a group of term neonates. We retrospectively searched all the data we had previously collected (and that has not

**Table 1.** Participant demographics.

Values given are median (lower quartile, upper quartile) or number (%). * Indicates missing data for one neonate.

| | Study 1 | Study 3 Control group | Study 3 Intervention group | Study 4 Control group | Study 4 Intervention group | Template validation |
|---|---|---|---|---|---|---|
| Number of neonates | 9 | 18 | 20 | 15 | 14 | 16 |
| Gestational age (GA) at birth (weeks) | 40.7 (40.3, 41) | 40 (37.1, 40.7) | 39.1 (37.1, 40.6) | 27.6 (25.6, 28.8) | 27.3 (26.3, 28.3) | 40.6 (40, 41) |
| Postmenstrual age (PMA) at time of study (weeks) | 41 (40.9, 41.7) | 40.5 (37.6, 40.9) | 39.5 (37.8, 41.1) | 38 (37.2, 39.4) | 37.2 (36.2, 38.1) | 40.7 (40.1, 41.4) |
| Postnatal age (PNA) at time of study (days) | 2 (2, 4) | 1.5 (1, 3.5) | 4 (2, 5) | 64 (59, 90) | 64 (62, 70.8) | 2 (1, 3) |
| Birthweight (g) | 4140 (3725, 4320) | 3675 (3021, 3881) | 3390 (3051, 3908) | 1040 (705, 1268) | 880 (708, 1031) | 3520 (3103, 3786) |
| Sex | | | | | | |
| Male | 4 (44) | 11 (61) | 9 (45) | 9 (60) | 9 (36) | 12 (75) |
| Female | 5 (56) | 7 (39) | 11 (55) | 6 (40) | 5 (64) | 4 (25) |
| Mode of delivery | | | | | | |
| NVD (normal vaginal delivery) | 2 (22) | 6 (33) | 7 (35) | 6 (40) | 3 (21.4) | 8 (50) |
| Assisted vaginal ventouse/forceps | 6 (67) | 3 (17) | 4 (20) | 0 (0) | 1 (7.1) | 4 (25) |
| Emergency C-section | 0 (0) | 6 (33) | 5 (25) | 7 (47) | 7 (50) | 4 (25) |
| Elective C-section | 1 (11) | 3 (17) | 4 (20) | 2 (13) | 3 (21.4) | 0 (0) |
| Apgar score at 1 min | 9 (8, 10) | 9 (8, 10) | 9 (7, 10) | 5 (4, 6)* | 7 (3, 8)* | 9 (9, 10) |
| Apgar score at 5 min | 10 (10, 10) | 10 (10, 10) | 10 (9, 10) | 8 (7, 9)* | 8 (7, 10)* | 10 (10, 10) |
| Estimated cumulative prior pain exposure | 4 (1, 4)* | 2 (2, 6) | 5 (3, 8)* | 31 (28, 451) | 78 (58, 219) | 0 (0, 0) |

been previously published) to identify any term neonates who had received a clinically required heel lance and experimental noxious stimuli on the same test occasion. We identified nine neonates studied between 2014 and 2015 (age range 39–42 weeks' gestational age) who had all received experimental noxious stimuli at a force of 64 mN. The magnitude of the noxious-evoked brain activity was characterised by projecting a previously described template of noxious-evoked brain activity (*Hartley et al., 2017*) (see 'Recording techniques anddata preparation' for further details) in response to each stimulus. The mean response to the experimental noxious stimulus in each neonate was correlated with their responses to the heel lance using a Pearson correlation test.

## Study 2: Simulating the effect of accounting for individual baseline sensitivity to reduce sample sizes

To investigate potential differences in the power achieved by accounting for individual baseline sensitivity at different sample sizes (for a given effect size and significance level), we simulated data sets. The code for these simulations is available from https://gitlab.com/paediatric_neuroimaging/simulating_power_nociceptive_sensitivity.git (*Cobo, 2021*; copy archived at swh:1:rev:1a465a40228c1c5ab5d33d4cbdf8cd99b29e9fcf). Each simulation consisted of a Control Group and an Intervention Group. Given a sample size of N per group, we first simulated N individual baseline sensitivity levels for each group by generating N uniform random numbers within the range of expected sensitivities. The minimum expected sensitivity was set as the minimum response to the experimental noxious stimuli in the data collected in Study 1, and the maximum expected response was set from multiplying the maximum of data collected in Study 1 by 3 (the expected increase in range from changing to an experimental noxious stimulus with a force of 128 mN from previously published data – Study 2 in *Hartley et al., 2017*, as using a force of 128 mN is expected to increase the signal-to-noise ratio). Thus, N individual baseline sensitivities, $x_i$, were generated per group with

$$x_i \in [0.15, 1.85], \ i \in \{1, \ldots, N\}$$

Responses to the clinical stimulus, $y_i$, were then simulated from these randomly generated individual baseline sensitives with:

$$y_i = 2.62 x_i - 0.75 + \varepsilon_i$$

where $\varepsilon_i$ is a noise term, and the values 2.62 and 0.75 were related to the line of best fit in Study 1 (the gradient of the line was reduced from Study 1 to account for the increase in the range of individual baseline sensitivities, that is, as the range of $x_i$ was increased from Study 1, the gradient was reduced so that the maximum value of $y_i$ was not higher than the expected response magnitude to a heel lance).

$\varepsilon_i$ was drawn from a random normal distribution with mean 0 and a standard deviation $\xi$. The standard deviation of residuals $\xi$ was set to be 0.37 in most simulations as this is the standard deviation of the residuals from Study 1 but was varied for the simulations in *Figure 2E*.

Finally, the responses to the clinical stimulus in the Intervention Group were reduced by a proportion according to the intervention effect. For most simulations, the intervention effect was set at 40% as this is considered a clinically meaningful effect size (*Lorenz et al., 1997*; *von Mohr et al., 2018*). However, varying levels of intervention effects were also investigated. We compared the responses in the Control Group and the Intervention Group with and without accounting for baseline sensitivity (see 'Statistical analysis').

For each value of N, 1000 Control and Intervention groups were simulated and the percentage of simulations where the group comparisons had p<0.05, that is, the power, was calculated. For simulations where the intervention effect or the noise level was varied, the minimum number of neonates required for a power of 95% was calculated by increasing the group size by 1 (note all data was re-simulated with each new sample size – so each simulated data set was fully independent) and calculating the power at each step until a power of 95% was achieved.

## Study 3: Testing the paradigm: a non-pharmacological pain-relieving intervention study

The aim of this study was to test the noxious-evoked baseline sensitivity paradigm using a gentle touch intervention of known effect in reducing the noxious-evoked brain activity following a clinically required heel lance (*Gursul et al., 2018*). The sample size required was obtained from the data simulated in Study 2. Assuming a 40% reduction in the magnitude of the brain activity in the Intervention Group compared with the Control Group, a sample size of 32 neonates (16/group) would be sufficient to achieve a power of 95%, when a neonate's baseline sensitivity is taken into account. Neonates were recruited in parallel to the Control or Invertenvion Group and the allocation of participants to the experimental groups was not random. A total of 40 neonates aged 35–42 weeks' postmenstrual age (PMA) were prospectively recruited to the study. EEG and electromyography (EMG) activities were recorded in response to experimental noxious stimuli (128 mN intensity) prior to a clinically required heel lance. The Intervention Group (n = 22) received gentle touch at an approximate rate of 3 cm/s for 10 s before heel lancing and the Control Group (n = 18) did not receive the gentle touch (all neonates received comfort measures including swaddling, non-nutritive sucking, or were held by parent). Gentle touch was not applied prior to the experimental stimuli. The gentle touch stimulus was provided by a brush stimulator (SENSELab Brush-05, Somedic) applied at a rate of approximately 3 cm/s for 10 s prior to heel lancing, across approximately 10 cm of the neonate's lower leg ipsilateral to the heel receiving the lance. Current evidence suggests that this rate (3 cm/s) optimally activates C-tactile fibres involved in the detection of pleasant touch (*Essick et al., 2010*; *Löken et al., 2009*; *Triscoli et al., 2014*) and our previous study demonstrated the efficacy of gentle touch to reduce brain-derived measures following a clinically required heel lance (*Gursul et al., 2018*). The experimenter was cued to apply the brushing velocity and noxious stimuli by following a computer visualisation coded using PsychoPy. There was an inter-stimulus interval of approximately 1 s between the end of the brush stimulation and the heel lance.

The magnitude of the noxious-evoked brain activity and the reflex withdrawal was obtained for each individual trial. Each individual neonates' baseline sensitivity was calculated as the mean response to the experimental noxious stimulus. EEG responses were rejected for gross movement artefacts. Following removal of neonates whose lances recording were rejected (n = 5), 118 out of 492 responses to experimental noxious stimuli were rejected from the EEG analysis. Individuals with seven or less responses to the experimental noxious stimuli were not included in the analysis (n = 4) as accurate estimates of baseline sensitivity could not be obtained. This left a total of 31 neonates (Control Group: n = 15; Intervention Group: n = 16) for the analysis. Similarly, EMG traces with noise and movement artefacts in the baseline period before the stimulus were rejected. Following removal of neonates whose lances recording were rejected (n = 7), 18 out of 459 responses to experimental noxious stimuli were rejected, leaving a total of 33 neonates (Control Group: n = 18; Intervention Group: n = 15) included in the EMG analysis. The individual baseline reflex sensitivity was calculated as the median reflex response to the experimental noxious stimulus.

## Study 4: A pharmacological analgesic study

Premature-born neonates aged 33–43 weeks' PMA and due to receive immunisations as inpatients in the neonatal unit were recruited for this study. Neonates received diphtheria, tetanus, acellular pertussis, polio, *Haemophilus influenzae* type b, hepatitis B (DTaP/IPV/Hib/HepB), meningococcal group B (MenB), and pneumococcal (PCV) intramuscular immunisations at 8 weeks' postnatal age, DTaP/IPV/Hib/HepB immunisations at 12 weeks' postnatal age, or DTaP/IPV/Hib/HepB, MenB, and PCV at 16 weeks' postnatal age. Thus, each neonate received one or three injections into one or both thighs. For the week 8 and 16 immunisations where MenB vaccine was due, oral paracetamol (15 mg/kg) was administered for the management of pyrexia, in line with the National Institute for Health and Care Excellence (NICE) and British National Formulary for Children (BNFc) guidelines for neonates born at less than 4 kg.

A total of 16 neonates (Control Group, *Figure 5A*) were recruited to the study before the clinical practice guidelines were updated in our local neonatal unit to administer paracetamol 1 hr prior to the MenB vaccine. Following the guideline change, 16 neonates were recruited (Intervention Group), and a subset of 11 of the 16 neonates in the Intervention Group also received experimental noxious

stimuli before and approximately 1 hr after paracetamol was administered (*Figure 5A*). The average time between paracetamol administration and the second set of experimental noxious stimuli in neonates in the Intervention Group subset was 60 min (range: 52–70 min). The average time between paracetamol administration and immunisation in neonates in the Intervention Group was 79 min (range: 65–117 min). Comfort techniques including swaddling or non-nutritive sucking were used during the immunisation procedures for neonates in both groups.

EEG was recorded continuously for the duration of the clinical procedure (i.e. immunisation). Needle insertion was time-locked to the EEG recordings using a high-speed video camera (220 frames per second; Firefly MV, Point Grey Research Inc) that was linked to the recording at the time of acquisition. The time of each individual stimulus was identified retrospectively from the video recordings as the first point of contact of the needle with the skin (*Hartley et al., 2014*; *Verriotis et al., 2016*). Due to technical difficulties (high-speed camera failure during set-up) and accidental deletion of a recording, two neonates were removed from the analysis. Recordings with poor video footage (for which the first point of contact of the needle with the skin was unidentifiable) were rejected from the analysis as well as traces with noise or movement artefact. A total of 29 neonates (Control Group: 15 neonates, 32 immunisations; Intervention Group: 14 neonates, 27 immunisations) were included in the final analysis, with nine neonates (18 immunisations) included in the Intervention Group subset. The magnitude of the noxious-evoked brain activity was identified in each individual trial, and for neonates in the subset of the Intervention Group baseline sensitivity was calculated as the neonate's mean response to the experimental noxious stimuli prior to paracetamol administration. This was compared to the neonate's mean response to the immunisations and mean response to the experimental noxious stimulus recorded 1 hr after paracetamol administration.

## Stimulation techniques/clinical procedures

### Experimental noxious stimuli

Experimental noxious stimuli (PinPrick, MRC Systems) of 64 mN intensity (Study 1) and 128 mN intensity (Studies 3 and 4) were applied to the plantar surface of the heel. The PinPrick applies a constant force that stimulates A∂ and C fibre peripheral nociceptors (*Magerl et al., 2001*), without piercing the skin. When applied to neonates at these forces, the stimulus does not cause behavioural distress or clinical concern (*Goksan et al., 2015*; *Hartley et al., 2017*; *Hartley et al., 2015*). Stimuli were applied in trains of 10–20 trials, with a minimum inter-stimulus interval of 10 s (the inter-stimulus interval was increased if necessary, to allow the neonate to settle). The 64 mN pinprick stimuli were time-locked to the EEG and EMG recordings using a high-speed video camera (220 frames per second; Firefly MV, Point Grey Research Inc) that was linked to the recordings at the time of acquisition. The time of each individual stimulus was identified retrospectively from the video recordings with a manual marker when the pinprick's barrel was first depressed (*Hartley et al., 2015*). The 128 mN pinprick was automatically time-locked to the EEG and EMG recordings using a contact trigger device (MRC Systems).

### Heel lance

Heel lances were applied only when necessary for blood sampling as part of the neonate's clinical care. Heel lances were automatically time-locked to the EEG and EMG recordings using an event detection interface (*Worley et al., 2012*). Comfort techniques including swaddling or non-nutritive sucking were used during the heel lance procedures.

## Recording techniques and data preparation

Electrophysiological activity was recorded from DC to 400 Hz using a SynAmps RT 64-channel headbox and amplifiers (Compumedics Neuroscan). CURRYscan7 neuroimaging suite (Compumedics Neuroscan) was used to record activity, with a sampling rate of 2000 Hz. EEG was recorded from eight locations on the scalp (Cz, CPz, C3, C4, Oz, FCz, T3, T4), with reference at Fz and ground at Fpz (forehead) according to the modified international 10–20 system. Preparation gel (Nuprep gel, D.O. Weaver and Co.) was used to gently clean the scalp with a cotton bud before disposable Ag/AgCl cup electrodes (Ambu Neuroline) were placed with conductive paste (Elefix EEG paste, Nihon Kohden). In Study 3, surface EMG was recorded from the limb ipsilateral to the site of stimulation.

Bipolar EMG electrodes (Ambu Neuroline 700 solid gel surface electrodes) were placed on the bicep femoris muscle.

EEG signals were filtered from 0.5 to 30 Hz with a notch filter at 50 Hz. Epochs were extracted 500 ms before the stimulus and 1000 ms after and were baseline-corrected to the pre-stimulus mean. Epochs were rejected if they contained gross movement artefact. Noxious-evoked brain activity was analysed at the Cz electrode for all trials (as this is the electrode site at which the maximal evoked response is observed; *Hartley et al., 2017*). The previously validated template of noxious-evoked brain activity was projected onto each individual trial in the time window of interest (400–700 ms after stimulation when the stimulus was applied to the foot, 300–600 ms after stimulation when the stimulus was applied to the thigh – see Methods to validate the template of noxious-evoked brain activity) providing a weight indicating the magnitude of the noxious-evoked brain activity (*Hartley et al., 2017*). Each individual trial was first Woody filtered in the time window of interest to achieve maximum correlation with the template, accounting for individual differences in the latency to the response. A maximum jitter of ±50 ms for the experimental noxious stimuli and ±100 ms for the heel lance and immunisations was used for the Woody filtering. In Study 4, additional variation in the latency of the response occurred from the use of the high-speed video camera. To account for this, responses were first Woody-filtered within a neonate to achieve maximum correlation with the within-subject average.

EMG signals were filtered 10–500 Hz, with a notch filter at 50 Hz and harmonics, and rectified. Epochs were extracted from 2 s prior to 4 s after the stimulus. Individual epochs were rejected due to movement artefact in the baseline period. The data was split into 250 ms windows and the root mean square (RMS) of the reflex signal was calculated in each window. The average RMS across the first four windows after the stimulus (first second after stimulation) was calculated as the magnitude of the reflex withdrawal.

## Methods to validate the template of noxious-evoked brain activity
### Accounting for latency differences when the stimuli is applied to different body sites

The template of noxious-evoked brain activity has previously been validated for experimental and clinical stimuli applied to the heel (*Hartley et al., 2017*). As other clinical interventions like immunisations are injected into the neonate's thigh, the latency to the brain activity response is expected to be shorter compared with stimuli applied to the foot. In an independent sample of 16 neonates aged 36–42 weeks' PMA (demographic details given in *Table 1*), we investigated the latency of the noxious-evoked brain activity following experimental noxious stimuli applied to the foot, thigh, and hand. A total of 10–12 experimental noxious stimuli (128 mN, inter-stimulus interval of at least 10 s) were applied to the neonate's hand (n = 16 neonates), foot (n = 16), and thigh (n = 10). The order of the stimulus location and the side were randomly selected by the research team before each test occasion (right = 7, left = 9). In seven studies, the EEG recordings were linked to a high-speed camera (Firefly MV, Point Grey Research Inc) to time-lock the experimental stimuli (*Hartley et al., 2015*). In the other nine studies, the stimuli were time-locked to the EEG recordings using a contact trigger device (MRC Systems).

The EEG signal was filtered 0.5–30 Hz with a notch filter at 50 Hz. Epochs were extracted 500 ms before the stimulus and 1000 ms after (total 1500 ms per epoch), and the traces were baseline-corrected to the pre-stimulus mean. Noxious-evoked brain activity was analysed at the Cz electrode for all the trials. Individual EEG epochs with artefacts were removed from the analysis and neonates with less than five trials on any individual locations were removed from the analysis for that location. The final foot, thigh, and hand analysis included 14, 8, and 14 neonates, respectively.

Data from all individual trials from all neonates were Woody-filtered in the 0–700 ms interval after the stimulus onset with a maximum shift of ±50 ms (aligning to the average of the data). The average response for each subject to the stimulus and the average background were calculated from the Woody-filtered data. Clusters of timepoints where the noxious stimulus was significantly different from background were identified using a nonparametric cluster analysis (*Maris and Oostenveld, 2007*). The cluster-based test statistic was calculated from 1000 random permutations of the data, and the threshold for cluster significance was set as the 97.5 percentile of the permuted data. The midpoint of the cluster was identified and the time window for the principal component analysis

(PCA) was taken as the 300 ms window about this midpoint, rounding to the nearest 100 ms, for each of the responses to stimuli applied to the hand, foot, and thigh separately. PCA was performed to identify the principal components (PCs) of the activity in response to the stimuli compared with the background activity (*Bromm and Scharein, 1982*; *Fabrizi et al., 2011*; *Hartley et al., 2016*; *Slater et al., 2010b*) and the PC weights compared between background and the stimulus response using a paired t-test. The first two PCs accounted for over 73% of the variance in the data across all stimulus conditions and were the only components tested. The PC in which the weights were significantly different in response to the noxious stimulus compared with background activity was selected as the noxious-evoked response. This PC and the previously described template of noxious-evoked brain activity (*Hartley et al., 2017*) were compared using correlation, to demonstrate the validity of using the template to identify noxious-evoked brain activity for stimuli applied to different body locations.

Consistent with previous studies (*Hartley et al., 2017*; *Hartley et al., 2015*), experimental noxious stimuli applied to the foot evoked a cluster of activity that was significantly different from background activity in the time window from 456 to 654 ms after the stimulus (p=0.014, cluster-corrected nonparametric test, *Figure 5—figure supplement 1A*). PCA applied in the time window 400–700 ms identified a representative waveform of the noxious-evoked response – the second PC had significantly higher weights following the noxious stimulation compared to background brain activity (p=0.035, *Figure 5—figure supplement 1B*) and was significantly correlated with the validated template of noxious-evoked brain activity (r = 0.97, p<0.001, *Figure 5—figure supplement 1C*). Experimental noxious stimuli applied to the hand evoked a cluster of activity that was significantly different from background activity in the time window 298–461 ms (p=0.01, *Figure 5—figure supplement 1D*; *Kasser et al., 2019*). PCA was applied in the time window 200–500 ms following stimulation; the weights of the second PC were significantly higher following the experimental noxious stimuli compared with background activity (p<0.001, *Figure 5—figure supplement 1E*) and this PC was highly correlated with the template of noxious-evoked brain activity (r = 0.97, p<0.001, *Figure 5—figure supplement 1F*).

Experimental noxious stimuli applied to the thigh evoked a cluster of activity that was significantly different from background activity in the time window 280–563 ms (p=0.002, *Figure 5—figure supplement 1G*). PCA was applied in the time window 300–600 ms following stimulation; the weights of the second PC were significantly higher following the experimental noxious stimuli compared with background activity (p<0.001, *Figure 5—figure supplement 1H*) and this PC was highly correlated with the template of noxious-evoked brain activity (r = 0.98, p<0.001, *Figure 5—figure supplement 1I*). Overall, the latencies of noxious-evoked brain activity are related to the physical distance of the stimulus location from the brain (*Figure 5—figure supplement 1J*) as expected. With immunisation applied to the thigh, the latency to the response is expected to be approximately 300 ms.

## Accounting for different modalities for use in studies of immunisation

To validate the suitability of the template (in the time window from 300 to 600 ms) to characterise immunisation-evoked brain activity, we compared the activity evoked by the immunisation with the background brain activity in the Control Group in Study 4 (n = 15) who did not receive paracetamol prior to immunisation. Clusters of timepoints where the noxious stimulus was significantly different from background were identified using a nonparametric cluster analysis, with 1000 random permutations of the data, to check that significant noxious-evoked activity was observed in the same time window as that observed in response to experimental noxious stimuli applied to the thigh. Immunisation-evoked activity was significantly different to background in the time window 416–594 ms (p=0.035, non-parametric cluster analysis, *Figure 5—figure supplement 2A*) following stimulation. PCA in the time window 300–600 ms identified the first PC weights (which accounted for 58% of the variance) as significantly higher following immunisation compared with background activity (p=0.004, *Figure 5—figure supplement 2B*) and this PC was highly correlated with the template of noxious-evoked brain activity (r = 0.97, p<0.001, *Figure 5—figure supplement 2C*). Therefore, the template of noxious-evoked brain activity, derived in an independent sample of neonates (*Hartley et al., 2017*), was considered appropriate to characterise response to immunisation and used in the subsequent analysis. An event-related potential with a similar waveform and latency has

been previously recorded in 1- and 2-month-old term-born neonates following immunisations (*Verriotis et al., 2016*).

## Statistical analysis

Statistical analysis was performed using MATLAB_R2020a (MathWorks). Linear associations were assessed using Pearson correlation tests in Studies 1, 3, and 4. Statistical significance (alpha<0.05) was assessed non-parametrically via permutation testing with 10,000 permutations using the PALM (permutation analysis of linear models) toolbox (*Winkler et al., 2014*). Group differences in Studies 2, 3, and 4 were assessed using linear regressions (unpaired two-sample t-tests, except for the differences in responses to the experimental noxious stimuli before and after paracetamol administration where a paired sample t-test was used). When using the paradigm to adjust for baseline sensitivity, we used the following linear regression model: $Y = b_0 + b_1X_1 + b_2X_2$, where $Y$ is the magnitude of the response to the clinical procedure, $b_0$ is the intercept, $X_1$ is the intervention, and $X_2$ is the baseline sensitivity. Without accounting for baseline sensitivity, the model used was $Y = b_0 + b_1X_1$. Statistical significance (alpha<0.05) in Studies 3 and 4 group comparisons was assessed non-parametrically via permutation testing with 10,000 permutations using PALM. The difference between the Intervention and Control Group in Study 4 was assessed using a linear mixed effects model, with subject and number of immunisations set as random effects. Two-sided tests were used for all statistical analyses with a significance level of 0.05.

## Acknowledgements

We would like to thank the neonates and their parents for taking part in this study. CH is a Wellcome Trust/Royal Society Sir Henry Dale Fellow (213486/Z/18/Z). Funding: The study was funded by the Wellcome Trust via a Senior Fellowship awarded to Rebeccah Slater (Grant number 207457/Z/17/Z).

## Additional information

### Funding

| Funder | Grant reference number | Author |
| --- | --- | --- |
| Wellcome Trust | Senior Fellowship Award | Rebeccah Slater |
| Wellcome Trust | 207457/Z/17/Z | Rebeccah Slater |

The funders had no role in study design, data collection and interpretation, or the decision to submit the work for publication.

### Author contributions

Maria M Cobo, Conceptualization, Data curation, Formal analysis, Validation, Investigation, Visualization, Methodology, Writing - original draft, Writing - review and editing; Caroline Hartley, Conceptualization, Data curation, Software, Formal analysis, Supervision, Investigation, Methodology, Writing - original draft, Writing - review and editing; Deniz Gursul, Conceptualization, Data curation, Formal analysis, Investigation, Visualization, Methodology, Writing - review and editing; Foteini Andritsou, Gabriela Schmidt Mellado, Ria Evans Fry, Gabrielle Green, Amy Hoskin, Data curation, Investigation, Writing - review and editing, Clinical oversight of the studies; Marianne van der Vaart, Miranda Buckle, Data curation, Investigation, Writing - review and editing; Luke Baxter, Data curation, Formal analysis, Writing - review and editing; Eugene P Duff, Formal analysis, Methodology, Writing - review and editing; Richard Rogers, Investigation, Writing - review and editing; Eleri Adams, Writing - review and editing, Clinical oversight of the studies; Fiona Moultrie, Conceptualization, Data curation, Supervision, Investigation, Writing - original draft, Writing - review and editing, Clinical oversight of the studies; Rebeccah Slater, Conceptualization, Resources, Supervision, Funding acquisition, Writing - original draft, Project administration, Writing - review and editing

## Author ORCIDs

Maria M Cobo ⓘD https://orcid.org/0000-0002-3961-1568
Caroline Hartley ⓘD https://orcid.org/0000-0002-7981-0836
Foteini Andritsou ⓘD http://orcid.org/0000-0002-4408-167X
Marianne van der Vaart ⓘD https://orcid.org/0000-0001-8628-8719
Gabriela Schmidt Mellado ⓘD https://orcid.org/0000-0002-5610-4357
Luke Baxter ⓘD http://orcid.org/0000-0001-9548-7162
Eugene P Duff ⓘD http://orcid.org/0000-0001-8795-5472
Ria Evans Fry ⓘD https://orcid.org/0000-0002-4062-6476
Fiona Moultrie ⓘD https://orcid.org/0000-0002-1431-791X
Rebeccah Slater ⓘD https://orcid.org/0000-0003-1595-4846

## Ethics

Human subjects: Studies were conducted in accordance with the Declaration of Helsinki and Good Clinical Practice guidelines. Ethical approval was obtained from the National Research Ethics Service (reference 12/SC/0447) and informed written parental consent was obtained prior to each study.

## Decision letter and Author response

Decision letter https://doi.org/10.7554/eLife.65266.sa1
Author response https://doi.org/10.7554/eLife.65266.sa2

# Additional files

## Supplementary files

• Transparent reporting form

## Data availability

Source data to produce Figures 2–5 are provided with the paper. The data that support the findings of this study are available upon reasonable request from the corresponding author. Due to ethical restrictions, we consider appropriate to monitor the access and usage of the data as it includes highly sensitive information. Data sharing requests should be directed to rebeccah.slater@paediatrics.ox.ac.uk. Code availability: The magnitude of noxious-evoked brain activity in response to the experimental noxious stimuli and clinically-required procedures was calculated using the template of noxious evoked brain activity previously validated for experimental and clinical stimuli and available from (Hartley et al., 2017). The code to perform simulations to compare the sample size needed to assess an intervention effect with and without taking into account individual nociceptive sensitivity presented in study 2 are available from https://gitlab.com/paediatric_neuroimaging/simulating_power_nociceptive_sensitivity.git (copy archived at https://archive.softwareheritage.org/swh:1:rev:1a465a40228c1c5ab5d33d4cbdf8cd99b29e9fcf).

The following datasets were generated:

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
