## [Decision Letter]

**Acceptance summary:**

This paper will be of great interest to pain researchers, neuroscientists, drug developers and clinicians. It introduces an EEG-based paradigm which can be used to determine infant's sensitivity to noxious stimuli and shows that this paradigm can help to reduce sample sizes in analgesic trials in infants.

**Decision letter after peer review:**

Thank you for submitting your article "Quantifying individual nociceptive sensitivity to optimise analgesic trials in infants" for consideration by *eLife*. Your article has been reviewed by 3 peer reviewers, including Markus Ploner as the Reviewing Editor and Reviewer #3, and the evaluation has been overseen by Christian Büchel as the Senior Editor. The following individual involved in review of your submission has agreed to reveal their identity: Dick Tibboel (Reviewer #1).

Essential revisions:

1. Absence of any information of individual patients with regards to whether the patients were naive for painful stimuli or not is striking and should be integrated both in the description of the patient groups as well as commented in the discussion part. Equally important is information about differences in post conceptual age of the different patient groups and the earlier experienced painful procedures.

2. The study claims that the magnitude of noxious-evoked brain activity elicited by a mild noxious stimulus is a novel measure of baseline nociceptive sensitivity. However, the reverse inference that neural activity observed during a clinically-painful procedure reflects nociceptive processing is not necessarily correct – given that previous studies have convincingly shown that virtually the same pattern of brain activity, both in EEG and fMRI, is observed when no pain (or nociception) is present. Thus, the nociceptive specificity of the findings should be critically discussed in light of recent literature and ongoing discussions of this topic. This discussion should also include a consideration of whether normalizing for amplitudes of brain responses to equally-intense stimuli of other modalities could be also effective to reduce sample sizes. Beyond, it should be explicitly stated that the present findings only refer to a particular type of phasic noxious stimuli but not necessarily to noxious stimuli of other types, e.g. slow-rising stimuli.

3. The study assumes that the inter-subject variability of the brain responses evoked by a noxious stimulus reflects pain *sensitivity*. However, this claim would require an assessment of the relationship between a gold standard measure of the inter-subject variability in pain reports or pain-related behavior and the magnitude of noxious-evoked brain activity. Thus, direct evidence that the brain responses to noxious stimuli reflect pain sensitivity is lacking. The more so as the gentle touch intervention in study 3 significantly reduced the magnitude of noxious-evoked brain responses but not of reflex withdrawal activity. This dissociation further questions the relationship between pain sensitivity and brain responses. Thus, the lack of a behavioral effect, the dissociation between behavioral effects and brain responses and the relationship between brain responses and pain should be critically discussed.

Reviewer #1:

This is an interesting study evaluating the relative role of individualized nociceptive sensitivity in newborns under a variety of conditions used as an argument to decrease the numbers needed for clinical trials. This is an important concept that could be taken into account in the planning of future analgesia related studies in this vulnerable group of patients.

This manuscript contains a number of interesting approaches of a group with a large experience in neonatal pain evaluation using objective measures such as EEG and imaging techniques like fMRI. Equally important is information about differences in post conceptual age of the different patient groups and the earlier experienced painful procedures. We know from the literature that the number is closely to ten a day. I feel that this kind of information is essential to reach significant conclusions and the generalizability of their findings for other conditions such as an experimental approach in the near future.

This manuscript contains a number of interesting approaches of a group with a large experience in neonatal pain evaluation using objective measures such as EEG and imaging techniques like fMRI. Taking into account individual nociceptive sensitivity is a long overlooked issue that refers to variability in pain thresholds which might blur the interpretation of the effect of both painfull stimuli as well as therapeutic effects in neonates admitted to ICU's. The four studies are a logical combination of a solid hypothesis generated approach which can be taken into account in future analgesic trials. There are a few comments that might enhance the quality of the manuscript which I like to see in a revised version. Absence of any information of individual patients with regards to whether the patients were naive for painful stimuli or not is striking and should be integrated both in the description of the patient groups as well as commented in the discussion part. Literature data are available on the change of pain thresholds in the course of NICU admission which might have an effect on the individual values that the authors obtained in their studies. Equally important is information about differences in post conceptual age of the different patient groups and the earlier experienced painfull procedures. We know from the literature that the number is closely to ten a day. I feel that this kind of information is essential to reach significant conclusions and the generalizability of their findings for other conditions such as an experimental approach in the near future.

Reviewer #2:

In this manuscript, Cobo and colleagues address the issue of measuring individual pain sensitivity in infants, and use this measure to improve the design of studies investigating the efficacy of analgesic treatments. The general question of measuring sensory sensitivity is of great interest in fundamental research, and the specific issue of sensitivity in pain experience has potential clinical relevance. For this reason, I accepted the task of assessing this manuscript with enthusiasm. However, when reading the manuscript, a fundamental flaw in the assumptions made by the authors together with a number of other major issues were apparent, as detailed below.

1. A central argument of the study by Cobo et al. is the claim that the magnitude of the noxious-evoked brain activity elicited by a mild noxious stimulus is a novel measure of baseline nociceptive sensitivity. This claim is based on the similarity between brain activity induced by a noxious stimulus and brain activity induced by a clinically painful procedure ("In Study 1, we demonstrate that the magnitude of noxious-evoked brain activity in response to an experimental stimulus correlates with the magnitude of brain activity evoked by the clinical procedure and thus reflects baseline nociceptive sensitivity"). The issue here is that the brain activity evoked by a clinically-painful procedure like an abrupt and short-lasting noxious stimulus (heel lance) is considered to reflect nociceptive processing. However, despite it is tempting to conclude that the brain activity observed during a clinically-painful procedure reflects the neural activity subserving nociceptive processing and the pain experience, this is not necessarily the case. In fact, there is a wealth of literature showing that the reverse inference consisting in concluding that the observed pattern of neural activity reflects nociceptive processing is incorrect – given that virtually the same pattern of brain activity, both in EEG and fMRI, is observed when no pain (or nociception) is present (for example, in response to equally salient, but never painful, auditory, tactile, and visual stimuli; see Iannetti and Hu Trends Neurosci 2016). All this literature, and the consequent ongoing discussion (e.g. Mouraux and Iannetti Brain 2018) is regrettably ignored by the authors.

The conclusion that the observed brain response reflects nociceptive processing and pain can be only drawn when a number of control stimuli, rigorously matched with the features that pain and nociception share with other sensory experiences (e.g. behavioural relevance, stimulus intensity, saliency), are used.

A final consideration is that the claim that the observed response reflects nociceptive processing should be substantiated by its obligatory nature – whereas there is plenty of evidence that slow-rising stimuli engaging the nociceptive system and causing very intense painful experience fail to evoke transient EEG responses like those used by Cobo et al. (see, for example, Colon, Liberati and Mouraux NeuroImage 2017).

2. Regardless of the previous issue, a second fundamental problem is that Cobo et al. assume that the inter-subject variability of the brain responses evoked by a noxious stimulus reflects pain *sensitivity*. What the author observe is simply a variability across subjects on the amplitude of the EEG response evoked by a transient stimulus – which is not a measure of nociceptive sensitivity. For this reason, claims like "we demonstrate that the magnitude of noxious-evoked brain activity in response to an experimental stimulus […] reflects baseline nociceptive sensitivity"are not granted. A measure of nociceptive sensitivity should (i) first assess the inter-subject variability in pain reports or, more relevant to the current study, pain-related behavior (the latter in cases in which the subjects cannot report their subjective percepts, like infants and experimental animals; see Peng et al. Pain 2017), and (ii) then assess the relationship between the magnitude of the noxious-evoked brain activity and the measures reflecting nociceptive sensitivity.

3. The authors claim that accounting for individual differences in "baseline nociceptive sensitivity" is able to reduce the sample size needed to assess the efficacy of analgesic interventions – study 2. However, it is unclear how this can be achieved. The authors suggest, as a possible strategy, to normalize for the brain activity evoked by the noxious stimulus, to reduce the individual differences (e.g. pages 20-21). If this is the case, it is imperative to test whether normalizing for the variability of a brain response evoked by equally-intense stimuli other sensory modalities (e.g., auditory) would be equally effective in reducing individual differences. Regrettably, this important control is lacking.

4. The authors mention that their paradigm could control for a vast array of known and unknown demographic and environmental factors that influence pain, and thereby result in inter-individual variability in responses, as well as potential experimental confounds (e.g., the difference in SNR, head circumference, and skull thickness). Does this mean that the measure they consider to reflect"baseline nociceptive sensitivity" would be also able to capture information associated with, for example, skull thickness? If this is the case, would auditory-evoked brain activity, which would be equally dependent on inter-subject variability of skull thickness, also reflect "baseline nociceptive sensitivity"?

5. It is of great interest that the authors mention that the noxious stimulation applied to the infants evokes a range of physiological responses, including reflex withdrawal and noxious-evoked brain activity. It is extremely important that the gentle touch intervention performed in study 3 did not significantly reduce the magnitude of the reflex withdrawal activity following heel lancing (while it reduces the magnitude of noxious-evoked brain activity). It is baffling that the authors conclude that the reduction of the brain activity evoked by noxious stimulation reflects the analgesic effect of the intervention, while the lack of reduction of pain-related behavior is ignored, and considered to be not important as a measure of the analgesic effect of the gentle touch. A solution would be to first validate the proposed paradigm in adults, who can more reliably report their subjective percept using a numerical rating scale. Regardless of the pain-related behavior issue, the lack of showing that the brain activity evoked by equally-intense auditory stimuli is not similarly dampened by the gentle touch makes it dangerous to conclude that the procedure is analgesic.

Reviewer #3:

The present study investigates whether EEG can be used to determine infant's individual sensitivity to noxious stimuli to eventually reduce sample sizes for analgesic trials. To this end, EEG responses have been recorded in response to low-intensity experimental noxious stimuli. Further analyses show that accounting for inter-individual differences of these EEG responses indeed allows for reducing sample sizes in analgesic trials. The approach is novel and innovative, the analysis straightforward, the results convincing and mostly support the conclusions of the study. A major strength of the study is that it includes a large group of infants and that the experiments are carefully designed and as standardized as possible in this particular population. Furthermore, the study represents a nice example of how basic research on the brain mechanisms of pain can be translated into clinical applications. The manuscript would, however, benefit from a critical discussion of two crucial points.

1. EEG responses to noxious stimuli mostly reflect supra-modal processes. EEG responses could therefore well reflect an individual's sensitivity to sensory stimuli in general rather than to noxious stimuli in particular. This doesn't invalidate the potential clinical use of the proposed paradigm but is important for the understanding of the underlying processes. Moreover, a possible lack of nociceptive specificity would mean that a paradigm assessing EEG responses to non-noxious stimuli could be equally powerful. Thus, the nociceptive specificity of the paradigm and the findings should be critically discussed with reference to the relevant literature on the specificity of EEG responses to noxious stimuli.

2. In study 3, the authors observed that the gentle touch intervention did not change reflex withdrawal activity in response to noxious stimuli. Thus, a direct indicator of the analgesic power of the touch intervention is lacking. The authors, however, found that the touch intervention reduced EEG responses to noxious stimuli. This dissociation between gentle touch effects on a measure of pain behavior and EEG responses questions the analgesic power of the intervention as well as a relationship between EEG responses on the one hand and pain and analgesia on the other hand. This crucial point should be explained and critically discussed.

---

## [Author Response]

Essential revisions:1. Absence of any information of individual patients with regards to whether the patients were naive for painful stimuli or not is striking and should be integrated both in the description of the patient groups as well as commented in the discussion part. Equally important is information about differences in post conceptual age of the different patient groups and the earlier experienced painful procedures.

We agree that data on prior pain exposure and age of the participants are important to provide contextual information and to define the limits for generalisability of the findings. Most studies were conducted in neonates born at term within the first 6 days after birth and with little prior pain exposure (number of previous painful procedures: study 1: median = 4, study 3: median = 4). Study 4 was conducted in prematurely born neonates aged 56-129 days at the time of the study and who had received a greater number of painful procedures (median = 58). This is information is highly relevant because larger noxious evoked potentials have been observed in infants who are born prematurely compared to those born at term (Slater et al., 2010) and heightened behavioural and motor responses have been described during clustered care in premature infants (Holsti et al., 2006, 2005). In addition, there is evidence to suggest an association between a greater number of painful procedures and lower pain-related behavioural responses within the first month of life (Grunau et al., 2001; Johnston and Stevens, 1996).

We have edited the manuscript to include this additional information.

(the page and line numbers provided are in reference to the Merged PDF file)

• Table 1 has been updated to include gestational age, postnatal age, and postmenstrual age, and prior pain exposure.

• The following text has been added to the Discussion (page 23, lines 443-446):

“The paradigm we present here likely accounts for multiple factors affecting noxious-evoked baseline sensitivity in neonates including potential effects from prior pain exposure during hospitalisation (Grunau et al., 2001; Johnston and Stevens, 1996) and prematurity (Slater et al., 2010).”

• The following text has been added to the Methods to describe how assessment of prior pain exposure was conducted (pages 31-32, lines 627-633):

“The estimate of cumulative prior pain exposure was quantified from each neonate’s clinical records as the total number of acute skin-breaking procedures (including heel lances, venepuncture, and intravenous cannulations) and aspirations (oropharyngeal or endotracheal) from time of birth to time of study (Hartley et al., 2016). These procedures were chosen based on a prospective epidemiology study describing the most commonly performed clinical procedures neonates are exposed to during hospitalisation (Carbajal et al., 2008).”

2. The study claims that the magnitude of noxious-evoked brain activity elicited by a mild noxious stimulus is a novel measure of baseline nociceptive sensitivity. However, the reverse inference that neural activity observed during a clinically-painful procedure reflects nociceptive processing is not necessarily correct – given that previous studies have convincingly shown that virtually the same pattern of brain activity, both in EEG and fMRI, is observed when no pain (or nociception) is present. Thus, the nociceptive specificity of the findings should be critically discussed in light of recent literature and ongoing discussions of this topic. This discussion should also include a consideration of whether normalizing for amplitudes of brain responses to equally-intense stimuli of other modalities could be also effective to reduce sample sizes. Beyond, it should be explicitly stated that the present findings only refer to a particular type of phasic noxious stimuli but not necessarily to noxious stimuli of other types, e.g. slow-rising stimuli.

We agree with the reviewers that our terminology was unintentionally too specific as our measure of noxious-evoked baseline sensitivity will also account for other factors unrelated to nociceptive processing (such as differences in the signal-to-noise ratio as described in the Discussion). Moreover, while the magnitude of noxious-evoked brain activity is likely to be closely related to the magnitude of the nociceptive input (Hartley et al., 2015), we do not provide evidence that the signal reflects activity that is nociceptive-specific.

We have changed the terminology throughout the manuscript and now use the term ‘noxious-evoked baseline sensitivity’ instead of ‘baseline nociceptive sensitivity’.

• The Introduction has been updated to clarify this definition (page 6, lines 100-109):

“The term ‘noxious-evoked baseline sensitivity’ is used to refer to differences in individual neonate’s noxious-evoked brain activity. […] In contrast to studies in adults which have shown similar patterns of activity evoked by both painful and non-painful stimuli (Mouraux et al., 2011), we have previously shown that the pattern of brain activity that we analyse here is not evoked by visual, auditory and tactile stimuli which evoke similar levels of physiological arousal (Hartley et al., 2017).”

We agree with the reviewers that other modalities, such as visual, auditory and tactile stimuli could be used to obtain a measure of baseline sensitivity, and this would account for some variance in the data. This is further supported by recent evidence that background resting-state brain activity is also predictive of an individual’s responses to noxious stimuli (Baxter et al., 2020). However, our objective was to develop a paradigm to optimise studies of analgesic efficacy. Therefore, the most appropriate stimulus to assess baseline sensitivity (which will likely account for the maximum variance in the data) is one with sensory modality matched to the subsequent clinically-required procedure. By recording baseline noxious-evoked activity prior to the painful procedure it is possible to (i) apply the stimuli to the same body site, (ii) record activity that is maximally evoked at the same electrode site and (iii) account for individual differences in somatosensory arousal. While in pain-related studies we consider it most appropriate to use noxious-evoked baseline sensitivity to adjust for inter-individual variability, if we were instead to conduct a vision-related studies we would consider it most appropriate to adjust for inter-individual variability using a visual stimulus.

• We have added the following text to the Discussion (pages 23-24, lines 451-466) to address this point:

“Responses to other modalities such as visual, auditory or tactile stimuli could be used to obtain a measure of baseline sensitivity, and background resting state brain activity is also predictive of individual noxious-evoked responses (Baxter et al., 2020). […] In contrast, if alternative studies considered other sensory modalities, for example visual processing, then a better measure of inter-individual baseline variability would be achieved using a visual stimulus.”

•We agree with the reviewers that our paradigm does not apply to all painful stimuli. We have a dded the following text to the Discussion (page 29, lines 590-595):

“In addition, while our paradigm is applicable to many of the most common acute somatic painful procedures which neonates are exposed to including heel lancing, cannulation and injections, this paradigm may not apply to many types of pain such as visceral pain, post-operative pain, longer procedures, such as retinopathy of prematurity screening, procedures with a slow-rising onset, or chronic pain.”

3. The study assumes that the inter-subject variability of the brain responses evoked by a noxious stimulus reflects pain sensitivity. However, this claim would require an assessment of the relationship between a gold standard measure of the inter-subject variability in pain reports or pain-related behavior and the magnitude of noxious-evoked brain activity. Thus, direct evidence that the brain responses to noxious stimuli reflect pain sensitivity is lacking. The more so as the gentle touch intervention in study 3 significantly reduced the magnitude of noxious-evoked brain responses but not of reflex withdrawal activity. This dissociation further questions the relationship between pain sensitivity and brain responses. Thus, the lack of a behavioral effect, the dissociation between behavioral effects and brain responses and the relationship between brain responses and pain should be critically discussed.

We agree with the reviewers that our terminology was too specific. Our paradigm is likely to account for variability in signals in non-neural features such as EEG set-up, skull thickness and head size, as well as inter-subject variability in brain responses evoked by the same intensity noxious input. The purpose of the study is to establish a paradigm that adjusts for both these neural and non-neural sources of variance in order to increase statistical power – not to establish the nociceptive-specificity or pain-specificity of the noxious-evoked baseline sensitivity signal. In the absence of verbalisation and a gold standard measure of pain in the neonatal population, our measure cannot be interpreted as variation in the amount of pain experienced by the individual infants. We do not know how much pain a non-verbal infant experiences. While clinical pain scores based on behavioural and autonomic responses are commonly used in infant pain literature, these measures lack specificity to pain or nociception and in the context of the study aims would not represent the main effect of interest.

With regards to the observation that reflex activity was not modulated by the gentle touch intervention we present three possible explanations:

(1) It is possible that we are underpowered to observe an effect, given the large variance observed in the reflex withdrawal responses and the weak correlation between the reflex to the experimental stimulus and to the clinical procedure. As stated in the Results, power calculations from the simulations conducted in Study 2 suggest we had power of only 17% to observe a significant difference between the magnitude of reflex withdrawal with the current sample size.

(2) It is possible that by applying the gentle touch to the leg immediately prior to the noxious procedure the baseline muscle activity and motor reflexes are being affected by this mechanical input, resulting in the larger correlation residuals that are observed in the Intervention Group compared to the Control Group.

(3) The gentle touch intervention may modulate the noxious-evoked brain activity (similar to distraction or other cognitive manipulations) but does not modulate the signal transmission of the nociceptive signal leading to the brain and therefore does not modulate the motor reflex activity. This is consistent with previous data (Gursul et al., 2018).

The International Association for the Study of Pain defines analgesia in terms of ‘pain perception’ which is a highly complex sensory and subjective emotional experience generated in the brain (IASP, 2020). Therefore, quantifying noxious-evoked brain activity may represent a better surrogate measure of pain, and a more sensitive marker of analgesic efficacy, compared with reflex signals generated by the spinal cord.

• The following text has been added to the Discussion to address this point (pages 25-26, lines 506-510):

“As pain perception is a highly complex sensory and subjective emotional experience generated in the brain (IASP, 2020) quantifying noxious-evoked brain activity may represent a better proxy pain measure, and a more sensitive marker of analgesic efficacy, compared with reflex signals generated by the spinal cord.”

In addition, we have changed the name of our study population from ‘infant’ to ‘neonate’ throughout the manuscript as this nomenclature is more appropriate for our study population.

Reviewer #1:This is an interesting study evaluating the relative role of individualized nociceptive sensitivity in newborns under a variety of conditions used as an argument to decrease the numbers needed for clinical trials. This is an important concept that could be taken into account in the planning of future analgesia related studies in this vulnerable group of patients.This manuscript contains a number of interesting approaches of a group with a large experience in neonatal pain evaluation using objective measures such as EEG and imaging techniques like fMRI. Equally important is information about differences in post conceptual age of the different patient groups and the earlier experienced painful procedures. We know from the literature that the number is closely to ten a day. I feel that this kind of information is essential to reach significant conclusions and the generalizability of their findings for other conditions such as an experimental approach in the near future.This manuscript contains a number of interesting approaches of a group with a large experience in neonatal pain evaluation using objective measures such as EEG and imaging techniques like fMRI. Taking into account individual nociceptive sensitivity is a long overlooked issue that refers to variability in pain thresholds which might blur the interpretation of the effect of both painfull stimuli as well as therapeutic effects in neonates admitted to ICU's. The four studies are a logical combination of a solid hypothesis generated approach which can be taken into account in future analgesic trials. There are a few comments that might enhance the quality of the manuscript which I like to see in a revised version. Absence of any information of individual patients with regards to whether the patients were naive for painful stimuli or not is striking and should be integrated both in the description of the patient groups as well as commented in the discussion part. Literature data are available on the change of pain thresholds in the course of NICU admission which might have an effect on the individual values that the authors obtained in their studies. Equally important is information about differences in post conceptual age of the different patient groups and the earlier experienced painfull procedures. We know from the literature that the number is closely to ten a day. I feel that this kind of information is essential to reach significant conclusions and the generalizability of their findings for other conditions such as an experimental approach in the near future.

Thank you for your comments. We agree with Reviewer 1 on the importance of describing the age of the neonates in the study and on considering prior pain exposure. This additional data has been added and discussed – see response to Essential revisions 1 above.

Reviewer #2:In this manuscript, Cobo and colleagues address the issue of measuring individual pain sensitivity in infants, and use this measure to improve the design of studies investigating the efficacy of analgesic treatments. The general question of measuring sensory sensitivity is of great interest in fundamental research, and the specific issue of sensitivity in pain experience has potential clinical relevance. For this reason, I accepted the task of assessing this manuscript with enthusiasm. However, when reading the manuscript, a fundamental flaw in the assumptions made by the authors together with a number of other major issues were apparent, as detailed below.1. A central argument of the study by Cobo et al. is the claim that the magnitude of the noxious-evoked brain activity elicited by a mild noxious stimulus is a novel measure of baseline nociceptive sensitivity. This claim is based on the similarity between brain activity induced by a noxious stimulus and brain activity induced by a clinically painful procedure ("In Study 1, we demonstrate that the magnitude of noxious-evoked brain activity in response to an experimental stimulus correlates with the magnitude of brain activity evoked by the clinical procedure and thus reflects baseline nociceptive sensitivity"). The issue here is that the brain activity evoked by a clinically-painful procedure like an abrupt and short-lasting noxious stimulus (heel lance) is considered to reflect nociceptive processing. However, despite it is tempting to conclude that the brain activity observed during a clinically-painful procedure reflects the neural activity subserving nociceptive processing and the pain experience, this is not necessarily the case. In fact, there is a wealth of literature showing that the reverse inference consisting in concluding that the observed pattern of neural activity reflects nociceptive processing is incorrect – given that virtually the same pattern of brain activity, both in EEG and fMRI, is observed when no pain (or nociception) is present (for example, in response to equally salient, but never painful, auditory, tactile, and visual stimuli; see Iannetti and Hu Trends Neurosci 2016). All this literature, and the consequent ongoing discussion (e.g. Mouraux and Iannetti Brain 2018) is regrettably ignored by the authors.The conclusion that the observed brain response reflects nociceptive processing and pain can be only drawn when a number of control stimuli, rigorously matched with the features that pain and nociception share with other sensory experiences (e.g. behavioural relevance, stimulus intensity, saliency), are used.A final consideration is that the claim that the observed response reflects nociceptive processing should be substantiated by its obligatory nature – whereas there is plenty of evidence that slow-rising stimuli engaging the nociceptive system and causing very intense painful experience fail to evoke transient EEG responses like those used by Cobo et al. (see, for example, Colon, Liberati and Mouraux NeuroImage 2017).

We thank the Reviewer for their detailed comments. Important points are raised by the reviewer which we have addressed in the revised version of the manuscript. As discussed in our reply to the Essential Revisions, we agree that our original terminology ‘baseline nociceptive sensitivity’ is too specific to nociception. We have standardised the terminology throughout the manuscript and now refer to this activity as ‘noxious-evoked baseline sensitivity’, as the brain activity is evoked by noxious stimulation (Magerl et al., 2001).

We agree that it is important to have an appropriate control stimulus that can be matched to the noxious stimuli if the aim of the study is to isolate the nociceptive components of the observed activity. This approach has been an important aspect of our previous work in infants, and in contrast to adults where similar patterns of activity are evoked by both painful and non-painful events, in infants the pattern of brain activity that we analyse here is not evoked by other equally arousing sensory stimuli (equivalent in terms of the evoked increase in heart rate) such as visual, auditory and tactile stimulation (Hartley et al., 2017). Nevertheless, the focus of this study is not to isolate a nociceptive-specific component of the response but to record an overall pattern of noxious-evoked brain activity which is more appropriate for adjusting for signal variance to gain statistical power and reduce sample sizes in neonatal analgesic studies.

2. Regardless of the previous issue, a second fundamental problem is that Cobo et al. assume that the inter-subject variability of the brain responses evoked by a noxious stimulus reflects pain sensitivity. What the author observe is simply a variability across subjects on the amplitude of the EEG response evoked by a transient stimulus – which is not a measure of nociceptive sensitivity. For this reason, claims like "we demonstrate that the magnitude of noxious-evoked brain activity in response to an experimental stimulus […] reflects baseline nociceptive sensitivity"are not granted. A measure of nociceptive sensitivity should (i) first assess the inter-subject variability in pain reports or, more relevant to the current study, pain-related behavior (the latter in cases in which the subjects cannot report their subjective percepts, like infants and experimental animals; see Peng et al. Pain 2017), and (ii) then assess the relationship between the magnitude of the noxious-evoked brain activity and the measures reflecting nociceptive sensitivity.

In the absence of language, we cannot know whether a non-verbal infant is in pain, and an alternative gold-standard pain measure in neonates does not exist. Noxious-evoked behavioural activity, reflex withdrawal activity, physiological responses (such as heart rate change) and brain activity do not directly measure neonatal pain. We agree with the reviewer that our measure does not reflect pain sensitivity. The inter-subject variability of brain responses evoked by the noxious stimulus will reflect differences in how the input signal is processed between participants due to both differences in their sensitivity to the stimuli and numerous other factors unrelated to pain and nociception, such as skull thickness. We have updated the terminology to refer to ‘noxious-evoked baseline sensitivity’ rather than ‘baseline nociceptive sensitivity’ throughout and have clarified in the Introduction exactly what we mean by this term. We have also clarified that the goal of the present study is not to establish the specificity of the noxious-evoked EEG signal to pain sensitivity or nociceptive sensitivity, but to describe a noxious-evoked baseline sensitivity signal that can be used to adjust for unwanted EEG signal variance ultimately reducing the sample size required for neonatal analgesia studies.

3. The authors claim that accounting for individual differences in "baseline nociceptive sensitivity" is able to reduce the sample size needed to assess the efficacy of analgesic interventions – study 2. However, it is unclear how this can be achieved. The authors suggest, as a possible strategy, to normalize for the brain activity evoked by the noxious stimulus, to reduce the individual differences (e.g. pages 20-21). If this is the case, it is imperative to test whether normalizing for the variability of a brain response evoked by equally-intense stimuli other sensory modalities (e.g., auditory) would be equally effective in reducing individual differences. Regrettably, this important control is lacking.

The noxious-evoked baseline sensitivity is recorded to increase the power and reduce the sample size needed to assess the efficacy of analgesic interventions; this is achieved by accounting for the variance explained by the baseline activity when comparing responses to the clinical procedure between the Control and Intervention groups (i.e. adding the baseline sensitivity as a covariate in the linear model). The details for the application of the experimental paradigm are shown in Figure 1. Noxious-evoked sensitivity paradigm explained.

The objective of this study was to develop a paradigm to optimise studies of analgesic efficacy. From a theoretical perspective, we think the most appropriate stimulus to assess baseline sensitivity for pain-related studies is one with sensory modality matched to the subsequent clinically-required painful procedure i.e. a noxious stimulus for subsequent painful stimulation, a visual stimulus for subsequent visual stimulation etc. Using baseline stimuli of different modality to the clinical procedure would require assessing the evoked signal at EEG electrodes (e.g. electrode Oz for visual-evoked activity) that differ from that subsequently used to measure the activity evoked by the painful procedure (i.e. electrode Cz). The more closely matched the setup is between the baseline assessment and clinical procedure assessment, the better the controlling for unwanted signal variance is likely to be.

4. The authors mention that their paradigm could control for a vast array of known and unknown demographic and environmental factors that influence pain, and thereby result in inter-individual variability in responses, as well as potential experimental confounds (e.g., the difference in SNR, head circumference, and skull thickness). Does this mean that the measure they consider to reflect"baseline nociceptive sensitivity" would be also able to capture information associated with, for example, skull thickness? If this is the case, would auditory-evoked brain activity, which would be equally dependent on inter-subject variability of skull thickness, also reflect "baseline nociceptive sensitivity"?

Our measure of noxious-evoked brain activity will be influenced by neural processes such as cognitive events, as well as non-neural features such as the EEG set up, skull thickness, and head size. However, the purpose of the paper is to establish a paradigm that adjusts for both these neural and non-neural sources of variance in order to reduce signal noise variance and increase statistical power. A noxious stimulus will provide a better baseline measure of all of these effects compared with an auditory stimulus. Less variance would be accounted for if we recorded auditory-evoked potentials.

5. It is of great interest that the authors mention that the noxious stimulation applied to the infants evokes a range of physiological responses, including reflex withdrawal and noxious-evoked brain activity. It is extremely important that the gentle touch intervention performed in study 3 did not significantly reduce the magnitude of the reflex withdrawal activity following heel lancing (while it reduces the magnitude of noxious-evoked brain activity). It is baffling that the authors conclude that the reduction of the brain activity evoked by noxious stimulation reflects the analgesic effect of the intervention, while the lack of reduction of pain-related behavior is ignored, and considered to be not important as a measure of the analgesic effect of the gentle touch. A solution would be to first validate the proposed paradigm in adults, who can more reliably report their subjective percept using a numerical rating scale. Regardless of the pain-related behavior issue, the lack of showing that the brain activity evoked by equally-intense auditory stimuli is not similarly dampened by the gentle touch makes it dangerous to conclude that the procedure is analgesic.

The observation that reflex activity was not modulated by the gentle touch intervention is discussed in detail in our reply to the Essential Revisions.

Reviewer #3:The present study investigates whether EEG can be used to determine infant's individual sensitivity to noxious stimuli to eventually reduce sample sizes for analgesic trials. To this end, EEG responses have been recorded in response to low-intensity experimental noxious stimuli. Further analyses show that accounting for inter-individual differences of these EEG responses indeed allows for reducing sample sizes in analgesic trials. The approach is novel and innovative, the analysis straightforward, the results convincing and mostly support the conclusions of the study. A major strength of the study is that it includes a large group of infants and that the experiments are carefully designed and as standardized as possible in this particular population. Furthermore, the study represents a nice example of how basic research on the brain mechanisms of pain can be translated into clinical applications. The manuscript would, however, benefit from a critical discussion of two crucial points.1. EEG responses to noxious stimuli mostly reflect supra-modal processes. EEG responses could therefore well reflect an individual's sensitivity to sensory stimuli in general rather than to noxious stimuli in particular. This doesn't invalidate the potential clinical use of the proposed paradigm but is important for the understanding of the underlying processes. Moreover, a possible lack of nociceptive specificity would mean that a paradigm assessing EEG responses to non-noxious stimuli could be equally powerful. Thus, the nociceptive specificity of the paradigm and the findings should be critically discussed with reference to the relevant literature on the specificity of EEG responses to noxious stimuli.

We would like to thank the Reviewer for their helpful comments. This point has been fully addressed in the Essential Revision section.

2. In study 3, the authors observed that the gentle touch intervention did not change reflex withdrawal activity in response to noxious stimuli. Thus, a direct indicator of the analgesic power of the touch intervention is lacking. The authors, however, found that the touch intervention reduced EEG responses to noxious stimuli. This dissociation between gentle touch effects on a measure of pain behavior and EEG responses questions the analgesic power of the intervention as well as a relationship between EEG responses on the one hand and pain and analgesia on the other hand. This crucial point should be explained and critically discussed.

The International Association for the Study of Pain defines analgesia in terms of pain perception which is a highly complex sensory and subjective emotional experience generated in the brain. It is important to note that although behavioural and autonomic responses are widely used in this population, these measures also lack specificity to pain (Slater et al., 2008), and do not allow for the discrimination between the sensory, cognitive, or emotional components of the experience. Therefore, quantifying noxious-evoked brain responses could represent a more objective and appropriate brain derived surrogate measure of pain compared with reflex signals generated at the level of the spinal cord. This point has now been explicitly outlined and discussed in our update manuscript.

References:

Baxter L, Moultrie F, Fitzgibbon S, Aspbury M, Mansfield R, Bastiani M, Rogers R, Jbabdi S, Duff E, Slater R. 2020. Functional and diffusion MRI reveal the functional and structural basis of infants’ noxious- evoked brain activity Author. Prepr (Version 1) available Res Sq. doi:10.21203/rs.3.rs-25860/v1

Carbajal R, Rousset A, Danan C, Coquery S, Nolent P, Ducrocq S, Saizou C, Lapillonne A, Granier M, Durand P, Lenclen R, Coursol A, Hubert P, Blanquat L de Saint, Boëlle P-Y, Annequin D, Cimerman P, Anand KJS, Bréart G. 2008. Epidemiology and Treatment of Painful Procedures in Neonates in Intensive Care Units. JAMA 300:60. doi:10.1001/jama.300.1.60

Grunau RE, Oberlander TF, Whitfield MF, Fitzgerald C, Lee SK, Background A. 2001. Low Birth Weight Neonates at 32 Weeks ’ Postconceptional Age. Pediatrics 107:105–112.

Gursul D, Goksan S, Hartley C, Mellado GS, Moultrie F, Hoskin A, Adams E, Hathway G, Walker S, McGlone F, Slater R. 2018. Stroking modulates noxious-evoked brain activity in human infants. Curr Biol 28:R1380–R1381. doi:10.1016/j.cub.2018.11.014

Hartley C, Duff EP, Green G, Mellado GS, Worley A, Rogers R, Slater R. 2017. Nociceptive brain activity as a measure of analgesic efficacy in infants. Sci Transl Med 9. doi:10.1126/scitranslmed.aah6122

Hartley C, Goksan S, Poorun R, Brotherhood K, Mellado GS, Moultrie F, Rogers R, Adams E, Slater R. 2015. The relationship between nociceptive brain activity, spinal reflex withdrawal and behaviour in newborn infants. Sci Rep 5. doi:10.1038/srep12519

Hartley C, Moultrie F, Gursul D, Hoskin A, Adams E, Rogers R, Slater R. 2016. Changing balance of spinal cord excitability and nociceptive brain activity in early human development. Curr Biol 26:1998–2002. doi:10.1016/j.cub.2016.05.054

Holsti L, Grunau RE, Oberlander TF, Whitfield MF. 2005. Prior pain induces heightened motor responses during clustered care in preterm infants in the NICU. Early Hum Dev 81:293–302. doi:10.1016/j.earlhumdev.2004.08.002

Holsti L, Grunau RE, Whitfield MF, Oberlander TF, Lindh V. 2006. Behavioral Responses to Pain Are Heightened After Clustered Care in Preterm Infants Born Between 30 and 32 Weeks Gestational Age. Clin J Pain 22:775–764.IASP. 2020. IASP Terminology – IASP. https://www.iasp-pain.org/Education/Content.aspx?ItemNumber=1698

Johnston CC, Stevens BJ. 1996. Experience in a neonatal intensive care unit affects pain response. Pediatrics 98:925–930.

Magerl W, Fuchs PN, Meyer RA, Treede R-D. 2001. Roles of capsaicin-insensitive nociceptors in cutaneous pain and secondary hyperalgesia. Brain 124:1754–1764. doi:10.1093/brain/124.9.1754

Mouraux A, Diukova A, Lee MC, Wise RG, Iannetti GD. 2011. A multisensory investigation of the functional significance of the “pain matrix.” Neuroimage 54:2237–2249. doi:10.1016/j.neuroimage.2010.09.084

Slater R, Cantarella A, Franck L, Meek J, Fitzgerald M. 2008. How well do clinical pain assessment tools reflect pain in infants? PLoS Med 5:0928–0933. doi:10.1371/journal.pmed.0050129

Slater R, Fabrizi L, Worley A, Meek J, Boyd S, Fitzgerald M. 2010. Premature infants display increased noxious-evoked neuronal activity in the brain compared to healthy age-matched term-born infants. Neuroimage 52:583–589. doi:10.1016/j.neuroimage.2010.04.253